# Reinforcement Learning from Human Feedback with Active Queries

**Kaixuan Ji**[*]                                                                *kaixuanji@cs.ucla.edu*
*Department of Computer Science*
*University of California, Los Angeles*

**Jiafan He**[*]                                                                *jiafanhe19@ucla.edu*
*Department of Computer Science*
*University of California, Los Angeles*

**Quanquan Gu**                                                                *qgu@cs.ucla.edu*
*Department of Computer Science*
*University of California, Los Angeles*

**Reviewed on OpenReview:** *https://openreview.net/forum?id=EScatQaRxz*

## Abstract

Aligning large language models (LLM) with human preference plays a key role in building modern generative models and can be achieved by reinforcement learning from human feedback (RLHF). Despite their superior performance, current RLHF approaches often require a large amount of human-labelled preference data, which is expensive to collect. In this paper, inspired by the success of active learning, we address this problem by proposing query-efficient RLHF methods. We first formalize the alignment problem as a contextual dueling bandit problem and design an active-query-based proximal policy optimization (APPO) algorithm with an $\widetilde{O}(d^2/\Delta)$ instance-dependent regret bound and an $\widetilde{O}(d^2/\Delta^2)$ query complexity, where $d$ is the dimension of feature space and $\Delta$ is the sub-optimality gap over all the contexts. We then propose ADPO, a practical version of our algorithm based on direct preference optimization (DPO) and apply it to fine-tuning LLMs. Our experiments show that ADPO, while only making about half of queries for human preference, matches the performance of DPO, establishing it as a data-efficient alternative to DPO. The codes are available at `https://github.com/jkx19/ActiveQuery`.

## 1 Introduction

Recent breakthroughs in large language models (LLM) significantly enhance the performances across a wide range of tasks, including common sense reasoning, world knowledge, reading comprehension and math problem solving (Jiang et al., 2023; Touvron et al., 2023; Chiang et al., 2023; Tunstall et al., 2023). In addition to the prominent capabilities of traditional natural language tasks (Gao et al., 2023a; Yuan et al., 2023; Han et al., 2023; Wei et al., 2023), they also demonstrate great potential in responding to human instructions (Ouyang et al., 2022). One key step towards building these models is aligning them with human preference, where reinforcement learning from human feedback (RLHF) (Fürnkranz et al., 2012; Casper et al., 2023; Ouyang et al., 2022; Ziegler et al., 2019; Christiano et al., 2017; Rafailov et al., 2024) is widely employed. The orthodox process of RLHF (Gao et al., 2023b; Munos et al., 2024) is described as follows. At each time, the human user prompts the LLM with an instruction. Subsequently, the model generates several candidate responses and queries the users for their preferences. Then, a reward model is trained on this preference data to mimic human evaluation. The language models are then updated using reinforcement learning (RL) algorithms such as Proximal Policy Optimization (PPO) (Schulman et al., 2017) to optimize responses that maximize the reward. However, PPO requires an additional reward model and online sampling from LLMs, which is computational inefficient. Alternatively, Direct Preference Optimization (DPO) (Rafailov et al.,

---

[*]Equal Contribution

2024) directly treats the language models themselves as the reward models and optimize the LLMs on the offline datasets. While its objective is mathematically equivalent to its canonical counterpart, it eliminates the requirement of additional reward modeling and online sampling.

Despite the notable success of RLHF in aligning language models with human preferences, its practical implementation often necessitates significant amounts of human-labeled preference data. For instance, the fine-tuning process of `zephyr-7b-beta` through RLHF relies on the utilization of a sizable 62k UltraCat-binarized dataset (Ding et al., 2023). The collection of such a substantial volume of human preference data is both costly and inefficient. Therefore, there exists a pressing need to develop query-efficient RLHF methods for effectively aligning large language models with human preferences.

Following recent theoretical advancements in RLHF (Xiong et al., 2023; Zhu et al., 2023; Sekhari et al., 2024), we formulate the RLHF problem as a contextual dueling bandit problem (Yue et al., 2012; Wu & Liu, 2016; Saha, 2021; Saha & Krishnamurthy, 2022; Saha & Gaillard, 2022; Wu et al., 2024; Di et al., 2024). In this setting, the learner proposes a pair of actions and receives noisy feedback regarding the preference between the dueling pair for each round. While numerous studies address regret minimization in dueling bandits, only a few works Wang et al. (2023); Zhan et al. (2023); Wu & Sun (2023); Sekhari et al. (2024) have considered query complexity during the learning process. However, their results either exhibit a linear dependency on the size of the action set $\mathcal{A}$, limiting the practical applicability of their methods, or fail to provide instance-dependent regret, thereby missing the opportunity to exploit favorable large-suboptimal-gap structures in RLHF.[1]

In this paper, we adopt the principles of active learning (Zhang & Oles, 2000; Hoi et al., 2006) to design a query-efficient algorithm, **A**ctive **P**roximal **P**olicy **O**ptimization (APPO) for linear contextual dueling bandits. In each round, APPO employs the maximum likelihood estimator (Di et al., 2024) to estimate the underlying parameter and constructs an optimistic estimator for the reward gap between different arms. Subsequently, APPO selects the best arms and estimates the uncertainty associated with the potential feedback. To reduce the query complexity, APPO selectively queries the dueling preference and updates the parameters only when the uncertainty of the observation exceeds a threshold.

We further extend APPO to direct preference optimization (DPO) (Rafailov et al., 2024) and introduce a novel query-efficient method, **A**ctive **D**irect **P**reference **O**ptimization (ADPO). Following the methodology of APPO, ADPO selectively queries human preference only for data where the model exhibits high uncertainty about the observation. For data where the model is less uncertain about, we employ the pseudo label predicted by the model to fine-tune the model itself. Our contributions are summarized as follows.

- We propose an active-learning based algorithm APPO for linear contextual dueling bandits. Theoretical analysis shows that our algorithm enjoys a constant instance-dependent regret $\widetilde{O}(d^2/\Delta)^2$, where $d$ is the dimension of the feature space, and $\Delta$ is the minimal sub-optimal gap. Meanwhile, our proposed algorithm only requires $\widetilde{O}(d^2/\Delta^2)$ queries in total $T$ rounds. Compared with previous instance-dependent regret bound $\widetilde{O}(A^2\beta^2 d/\Delta)$ achieved by Sekhari et al. (2024)[3], where $A$ is the size of the action space, our regret bound is independent on the size of action space $A$, which is more favorable in practice.

- We propose an active-learning-based DPO method named ADPO. We apply our method to train `zephyr-7b-beta` on Ultrafeedback-binarized dataset (Ding et al., 2023) and `zephyr-7b-gemma` on dpo-mix-7k dataset. Our experiment shows that while ADPO only make less than half numbers of queries, the model trained by ADPO achieves a comparable or better performance than DPO on our selected benckmarks including MT-Bench (Zheng et al., 2024) and AlpacaEval 2.0.

**Notation** We employ $[n]$ to denote the set $\{1, \ldots, n\}$. In this work, we use lowercase letters to represent scalars, and denote vectors and matrices by lower and uppercase boldface letters respectively. Given a vector $\mathbf{x} \in \mathbb{R}^d$, we denote the vector's $L_2$-norm by $\|\mathbf{x}\|_2$. We further define $\|\mathbf{x}\|_{\boldsymbol{\Sigma}} = \sqrt{\mathbf{x}^\top \boldsymbol{\Sigma} \mathbf{x}}$ given a positive semidefinite matrix $\boldsymbol{\Sigma} \in \mathbb{R}^{d \times d}$. We use standard asymptotic notations $O(\cdot), \Omega(\cdot), \Theta(\cdot)$, and further use $\widetilde{O}(\cdot)$ to hide logarithmic factors other than the number of rounds $T$. We use $\mathbb{1}\{\cdot\}$ denote the indicator function.

---

[1]For instance, in the Ultrafeedback-binarized dataset, the minimal reward gap is 0.5, within a range of 0 to 10. This occurs because users typically cannot perceive subtle quality differences between responses.

[2]We use notation $\widetilde{\mathcal{O}}(\cdot)$ to hide the log factor other than number of rounds $T$

[3]In our work, we only focused on the regret of one selected action, which is slightly different from the two-arm regret in Sekhari et al. (2024). See Section 3 for further discussion.

## 2 Related Work

**Reinforcement Learning from Human Feedback** Learning from human preference data dates back to Wirth et al. (2017); Christiano et al. (2017) and is recently popularized by generative language models (Achiam et al., 2023; Touvron et al., 2023). This procedure usually takes place after supervised finetuning (SFT). The canonical procedure of aligning with human preference includes two stages: reward modeling and reward maximization (Ouyang et al., 2022; Bai et al., 2022; Munos et al., 2024). Another approach is direct preference optimization (DPO) (Rafailov et al., 2024), which treats the generative models directly as reward models and trains them on preference data. Compared with the first approach, DPO simplifies the aligning process while maintaining its effectiveness. However, both paradigms require a large amount of human preference data. In this work, we follow the DPO approach and study its query-efficient modification. The empirical success of RLHF also prompts a series of theoretical works, with a predominant focus on the reward maximization stage, modeling this process as learning a dueling bandit (Zhu et al., 2023; Xiong et al., 2023; Sekhari et al., 2024). Among these works, Wang et al. (2023); Zhan et al. (2023); Wu & Sun (2023); Sekhari et al. (2024) stand out for considering query complexity in the learning process. However, Wang et al. (2023); Zhan et al. (2023); Wu & Sun (2023) focus either on worst-case regret bounds or the sample complexity to identify an $\epsilon$-optimal policy, but fail to provide instance-dependent guarantees (Further discussion is deferred to Appendix B). Only Sekhari et al. (2024) offer an instance-dependent analysis; however, their upper bound is $\widetilde{O}(A^2\beta^2 d/\Delta)$, which depends on the size of the action set $A$, limiting the practical applicability of their algorithm. Compared to this work, we provide an instance-dependent regret guarantee without dependency on the action space. Furthermore, based on APPO, we derive a practical algorithm, ADPO, which we empirically verify to demonstrate its superiority. We also notice two concurrent works that incorporate active learning with DPO. Mehta et al. (2023) incorporate active learning to DPO and use the variance of log-probabilities under different dropouts as the uncertainty estimator, which is inefficient in practice. Muldrew et al. (2024) also proposed an active learning-based alternative of DPO and leverage reward difference as uncertainty estimator. However, their approach does not involve pseudo labels, which is a key component of our approach.

**Active Learning** To mitigate the curse of label complexity, active learning serves as a valuable approach in supervised learning . The first line of work is pool-based active learning (Zhang & Oles, 2000; Hoi et al., 2006; Gu et al., 2012; 2014; Citovsky et al., 2021). In pool-based active learning, instead of acquiring labels for the entire dataset, the learner strategically selects a batch of the most informative data at each step and exclusively queries labels for this selected data batch. The learner then employs this labeled data batch to update the model. Subsequently, guided by the enhanced model, the learner queries another mini-batch of labels and continues the training process. These steps are iteratively repeated until the model achieves the desired performance level. The strategic selection of informative data significantly reduces the label complexity for supervised learning. The label complexity of pool-based active learning has been extensively studied by Dasgupta (2005); Dasgupta et al. (2005); Balcan et al. (2006; 2007); Hanneke & Yang (2015); Gentile et al. (2022). This strategy has also been widely applied in tasks like robotics learning (Akrour et al., 2012; Biyik & Palan, 2019; Wilde et al., 2020). On the other hand, selective sampling (a.k.a., online active learning) (Cesa-Bianchi et al., 2005; 2006; 2009; Hanneke & Yang, 2021) is a learning framework that integrates online learning and active learning. In this framework, the algorithm sequentially observes different examples and determines whether to collect the label for the observed example. In reinforcement learning, there are also lines of works focusing on the application of active learning. On theoretical side, Schulze & Evans (2018); Krueger et al. (2020); Tucker et al. (2023) focuses on active reinforcement learning and directly integrates the query cost into the received reward. Krueger et al. (2020) laid the groundwork for active reinforcement learning by introducing a cost $c$ associated with each reward observation and evaluated various heuristic algorithms for active reinforcement learning. Recently, Tucker et al. (2023) studied the multi-arm bandit problem with costly reward observation. Their work not only suggests empirical advantages but also proves an $O(T^{2/3})$ regret guarantee. On the application side, there are also lines of works apply selective sampling to specific circumstance of RLHF in robotics (Lee et al., 2021a;b; Liang et al., 2022).

## 3 Preliminaries

In this work, we formulate the RLHF problem as a contextual dueling bandit problem (Saha, 2021; Di et al., 2024). We assume a context set $\mathcal{X}$, and at the beginning of each round, a contextual variable $x_t$ is i.i.d generated from the context set $\mathcal{X}$ with the distribution $\mathcal{D}$. Based on the context $x_t$, the learner then chooses

two actions $y_t^1, y_t^2$ from the action space $\mathcal{A}$ and determines whether to query the environment for preferences between these actions. If affirmative, the environment generates the preference feedback $o_t$ with the following probability $\mathbb{P}(o_t = 1|x_t, y_t^1, y_t^2) = \sigma(r(x_t, y_t^1) - r(x_t, y_t^2))$, where $\sigma(\cdot) : \mathbb{R} \to [0, 1]$ is the link function and $r(\cdot, \cdot)$ is the reward model.

We consider a linear reward model, e.g., $r(x, y) = \langle \boldsymbol{\theta}^*, \boldsymbol{\phi}(x, y) \rangle$, where $\boldsymbol{\theta}^* \in \mathbb{R}^d$ and $\boldsymbol{\phi} : \mathcal{X} \times \mathcal{A} \to \mathbb{R}^d$ is a known feature mapping. For the sake of simplicity, we use $\boldsymbol{\phi}_t^1, \boldsymbol{\phi}_t^2$ to denote $\boldsymbol{\phi}(x_t, y_t^1), \boldsymbol{\phi}(x_t, y_t^2)$. Additionally, we assume the norm of the feature mapping $\boldsymbol{\phi}$ and the underlying vector $\boldsymbol{\theta}^*$ are bounded.

**Assumption 3.1.** The linear contextual dueling bandit satisfies the following conditions:

- For any contextual $x \in \mathcal{X}$ and action $y \in \mathcal{A}$, we have $\|\boldsymbol{\phi}(x, y)\|_2 \leq L/2$ and $r(x, y) \leq 1$.

- For the unknown environment parameter $\boldsymbol{\theta}^*$, it satisfies $\|\boldsymbol{\theta}^*\|_2 \leq B$.

For the link function $\sigma$, we make the following assumption, which is commonly employed in the study of generalized linear contextual bandits (Filippi et al., 2010; Di et al., 2024).

**Assumption 3.2.** The link function $\sigma$ is differentiable and the corresponding first derivative satisfied $\kappa_\sigma \leq \dot{\sigma}(\cdot)$, where $\kappa_\sigma > 0$ is a known constant.

The learning objective is to minimize the cumulative regret defined as:

$$\text{Regret}(T) = \sum_{t=1}^{T} r^*(x_t) - r(x_t, y_t^1),$$

where $r^*(x_t) = r^*(x_t, y_t^*) = \max_{y \in \mathcal{A}} r^*(x_t, y)$ stands for the largest possible reward in context $x_t$. It is worth noting that prior works (Di et al., 2024; Saha & Krishnamurthy, 2022; Sekhari et al., 2024) in dueling bandits often define the regret on both action $y_t^1$ and $y_t^2$. However, in the context of RLHF, the model generates multiple candidate responses, and users will choose the most preferable response from the available options. Under this circumstance, sub-optimality is only associated with the selected response. Therefore, we choose the regret defined only on action $y_t^1$.

To quantify the cost of collecting human-labeled data, we introduce the concept of query complexity $\text{Query}(T)$ for an algorithm, which is the total number of data pairs that require human feedback for preference across the first $T$ rounds. Note that while some prior work (Tucker et al., 2023) counts the cost of requesting for human feedback together with the cost paid for taking certain action, in our approach, we distinguish between regret and query complexity as two separate performance metrics for an algorithm.

In addition, we consider the minimal sub-optimality gap (Simchowitz & Jamieson, 2019; Yang et al., 2021; He et al., 2021), which characterizes the difficulty of the bandit problem.

**Definition 3.3** (Minimal sub-optimality gap)**.** For each context $x \in \mathcal{X}$ and action $y \in \mathcal{A}$, the sub-optimality gap $\Delta(x, y)$ and the minimal gap $\Delta$ are defined as

$$\Delta(x, y) = r^*(x) - r(x, y), \ \Delta = \min_{x \in \mathcal{X}, y \in \mathcal{A}} \{\Delta(x, y) : \Delta(x, y) \neq 0\}.$$

In general, a larger sub-optimality gap $\Delta$ between action $y$ and the optimal action $y^*$ implies that it is easier to distinguish between these actions and results in a lower cumulative regret. Conversely, a task with a smaller gap $\Delta$ indicates that it is more challenging to make such a distinction, leading to a larger regret. In this paper, we assume the minimal sub-optimality gap is strictly positive.

**Assumption 3.4.** The minimal sub-optimality gap is strictly positive, i.e., $\Delta > 0$.

**Remark 3.5.** In the context of RLHF, given a prompt, the minimal sub-optimality gap $\Delta$ represents the uniform gap between the best answers and the sub-optimal answers, where the optimal answers (or arms in the context of bandits) might not be unique (See Definition 3.3). Typically, for arms with sub-optimality close to 0, it is difficult for humans to discern a quality difference between them. Under this situation, we can roughly consider these arms also as optimal arms (optimal arms may not be unique) and only consider the gap between sub-optimal arms and these optimal arms. Therefore, Assumption 3.4 is mild in the context of RLHF.

---

**Algorithm 1** Active Proximal Policy Optimization (APPO)

---

**Require:** Regularization parameter $\lambda > 0$, and $B$, an upper bound on the $\ell_2$-norm of $\boldsymbol{\theta}^*$, confidence radius $\beta$, uncertainty threshold $\Gamma > 0$, learning rate $\eta$

1: Set initial policy $\pi_1(\cdot|\cdot)$ as uniform distribution over the action set $\mathcal{A}$, $\boldsymbol{\Sigma}_0 \leftarrow \lambda\mathbf{I}$, $\mathcal{C}_0 = \emptyset$
2: **for** $t = 1, \ldots, T$ **do**
3:     Compute the MLE $\widehat{\boldsymbol{\theta}}_t$ as in (4.1) and observe $\mathcal{A}$, select $y_t^2 \sim \text{Uniform}(\mathcal{A})$
4:     Compute $\widehat{D}_t(x_t, y) = \min\{\langle\widehat{\boldsymbol{\theta}}_t, \boldsymbol{\phi}(x_t, y) - \boldsymbol{\phi}_t^2\rangle + \beta\|\boldsymbol{\phi}(x_t, y) - \boldsymbol{\phi}_t^2\|_{\boldsymbol{\Sigma}_{t-1}^{-1}}, 1\}$
5:     Choose $y_t^1 = \text{argmax}_y \widehat{D}_t(x_t, y)$
6:     **if** $\|\boldsymbol{\phi}_t^1 - \boldsymbol{\phi}_t^2\|_{\boldsymbol{\Sigma}_{t-1}^{-1}} \leq \Gamma$ **then**
7:         Keep $\boldsymbol{\Sigma}_t = \boldsymbol{\Sigma}_{t-1}$, $\pi_{t+1}(a|s) = \pi_t(a|s)$ and $\mathcal{C}_t = \mathcal{C}_{t-1}$
8:     **else**
9:         Sample $y_t^1 \sim \pi_t(\cdot|s_t)$, query for the preference and observe $o_t$
10:        Update $\boldsymbol{\Sigma}_t = \boldsymbol{\Sigma}_{t-1} + (\boldsymbol{\phi}_t^1 - \boldsymbol{\phi}_t^2)(\boldsymbol{\phi}_t^1 - \boldsymbol{\phi}_t^2)^\top$ and $\mathcal{C}_t = \mathcal{C}_{t-1} \cup \{t\}$
11:        Update $\pi_{t+1}(y|x) \propto \pi_t(y|x)\exp\left(\eta\widehat{D}_t(y,x)\right)$
12:     **end if**
13: **end for**

---

## 4 Algorithm

In this section, we introduce our proposed query-efficient method for aligning LLMs. The main algorithm is illustrated in Algorithm 1. At a high level, the algorithm leverages the uncertainty-aware query criterion (Zhang et al., 2023) to issue queries and employs Optimistic Proximal Policy Optimization (OPPO) (Cai et al., 2020; He et al., 2022a) for policy updates. In the sequel, we introduce the key parts of the proposed algorithm.

**Regularized MLE Estimator** For each round $t \in [T]$, we construct the regularized MLE estimator (Filippi et al., 2010; Li et al., 2017) of parameter $\boldsymbol{\theta}^*$ by solving the following equation:

$$\lambda\boldsymbol{\theta} + \sum_{\tau \in \mathcal{C}_{t-1}}\left[o_\tau - \sigma\left(\langle\boldsymbol{\theta}, \boldsymbol{\phi}_\tau^1 - \boldsymbol{\phi}_\tau^2\rangle\right)\right](\boldsymbol{\phi}_\tau^1 - \boldsymbol{\phi}_\tau^2) = \mathbf{0}, \tag{4.1}$$

where $\mathcal{C}_t$ denotes the set of rounds up to the $t$-th round for which the preference label is required. Compared with previous work on linear dueling bandits (Saha, 2021; Di et al., 2024), here we only requires part of the human-labelled preference. We construct the MLE estimator with only rounds $\tau \in \mathcal{C}_t$. In addition, the estimation error between $\widehat{\boldsymbol{\theta}}_t$ and $\boldsymbol{\theta}^*$ satisfies

$$\|\boldsymbol{\theta}^* - \widehat{\boldsymbol{\theta}}_t\|_{\boldsymbol{\Sigma}_{t-1}} \leq \widetilde{O}\left(\sqrt{d\log|\mathcal{C}_t|}/\kappa_\sigma\right).$$

After constructing the estimator $\widehat{\boldsymbol{\theta}}_t$, the agent first selects a baseline action $y_t^2$ and compares each action $y \in \mathcal{A}$ with the baseline action $y_t^2$. For simplicity, we denote $D_t(x_t, y) = \langle\boldsymbol{\theta}^*, \boldsymbol{\phi}(x_t, y) - \boldsymbol{\phi}_t^2\rangle$ as the reward gap between $y$ and action $y_t^2$. Then, we construct an optimistic estimator $\widehat{D}_t$ for the reward gap with linear function approximation and Upper Confidence Bound (UCB) bonus, i.e.,

$$\widehat{D}_t(x_t, y) = \min\{\langle\widehat{\boldsymbol{\theta}}_t, \boldsymbol{\phi}(x_t, y) - \boldsymbol{\phi}_t^2\rangle + \beta\|\boldsymbol{\phi}(x_t, y) - \boldsymbol{\phi}_t^2\|_{\boldsymbol{\Sigma}_{t-1}^{-1}}, 1\}.$$

Here we truncate the estimation since the true reward is in $[0, 1]$ and therefore their difference is bounded by 1. With the help of UCB bonus, we can show that our estimated reward gap $\widehat{D}_t$ is an upper bound of the true reward gap $D_t$.

**Uncertainty-Aware Query Criterion** To mitigate the expensive costs from collecting human feedback, we introduce the uncertainty-based criterion (Line 6) (Zhang et al., 2023) to decide whether a pair of action $y_t^1$ and $y_t^2$ requires human labeling. Intuitively speaking, the UCB bonus $\beta\|\boldsymbol{\phi}_t^1 - \boldsymbol{\phi}_t^2\|_{\boldsymbol{\Sigma}_{t-1}^{-1}}$ captures the uncertainty associated with the preference feedback $o_t$. Similar criterion has also been used in corruption-robust linear contextual bandits (He et al., 2022b) and nearly minimax optimal algorithms for learning linear (mixture) Markov decision processes (Zhou & Gu, 2022; He et al., 2023; Zhao et al., 2023), where $\beta\|\boldsymbol{\phi}\|_{\boldsymbol{\Sigma}_{t-1}^{-1}}$ represents the uncertainty of certain action. For the action pair $(y_t^1, y_t^2)$ with low uncertainty, where the observation

is nearly known and provides minimal information, we select the action $y_t^1, y_t^2$ without querying human preference feedback. In this situation, the policy $\pi(\cdot|\cdot)$ remains unchanged as there is no observation in this round. By employing the uncertainty-based data selection rule, we will later prove that the query complexity is bounded.

**Proximal Policy Optimization** In cases where the action pair $(y_t^1, y_t^2)$ exhibits high uncertainty and the uncertainty-aware query criterion is triggered, the agent resample the action $y_t^1$ from policy $\pi_t$ and queries human feedback for the duel $y_t^1, y_t^2$. Upon observing the preference $o_t$, this round is then added to the dataset $\mathcal{C}_t$. Subsequently, the policy $\pi_{t+1}$ is updated using the Optimistic Proximal Policy Optimization (OPPO) method (Cai et al., 2020; He et al., 2022a), i.e.,

$$\pi_{t+1}(y|x) \propto \pi_t(y|x) \exp\left(\eta \widehat{D}_t(y, x)\right).$$

In an extreme case where the uncertainty threshold $\Gamma$ is chosen to be 0, the uncertainty-aware query criterion will always be triggered. Under this situation, Algorithm 1 will query the human-labeled preference for each duel $(y_t^1, y_t^2)$, and Algorithm 1 will degenerate to the dueling bandit version of OPPO (Cai et al., 2020). Under this situation, Algorithm 1 enjoys $\widetilde{O}(d\sqrt{T})$ regret while having a linear query complexity with respect to the number of rounds $T$.

# 5 Theoretical Analysis

In this section, we present our main theoretical results.

**Theorem 5.1.** Let $\Delta$ be the minimal sub-optimal gap in Assumption 3.4. If we set the parameters $\Gamma = \widetilde{O}(\Delta/\sqrt{d})$, $\lambda = B^{-2}$, $\eta = \widetilde{O}(\sqrt{\Gamma^2 \log \mathcal{A}/d})$, and $\beta = \widetilde{O}(\sqrt{d}/\kappa_\sigma)$ in Algorithm 1, then with probability at least $1 - \delta$, the regret for Algorithm 1 across the first $T$ rounds is upper bounded by

$$\text{Regret}(T) = \widetilde{O}(d^2/\Delta).$$

In addition, the query complexity of Algorithm 1 is upper bounded by:

$$\text{Query}(T) = |\mathcal{C}_T| = \widetilde{O}(d^2/\Delta^2).$$

**Remark 5.2.** Theorem 5.1 suggests that our algorithm achieves a constant level of regret and query complexity respect to the number of rounds $T$. In theory, our algorithm requires a prior knowledge of the sub-optimal gap $\Delta$. In practice where $\Delta$ is unknown, the learner can set the parameter $\Delta$ via grid search process.

**Remark 5.3.** In comparison to the instance-dependent regret $\widetilde{O}(A^2 d/\Delta)$ obtained by the AURORA algorithm (Sekhari et al., 2024)[4], our algorithm's regret eliminates the dependency of the action space $A$. Moreover, we achieve an improvement in the query complexity by a factor of $A^3$.

# 6 Practical Algorithm

In this section, we introduce a practical version of our proposed algorithm based on DPO (Rafailov et al., 2024) and the resulting algorithm is named as Active Direct Preference Optimization (ADPO) and summarized in Algorithm 2. At a high level, our proposed method follows the basic idea of Algorithm 1 and sets an uncertainty threshold to filter out informative training samples. However, adapting our algorithm to neural network training requires several key modifications as below.

**Direct Preference Optimization** We follow the framework of DPO Rafailov et al. (2024) for policy optimization. In detail, we consider the Bradley-Terry (BT) model (Bradley & Terry, 1952), which corresponds to $\sigma(x) = 1/(1 + e^{-x})$. In RLHF, the objective is to maximize the expected reward regularized by the Kullback-Leibler (KL) divergence from the reference policy $\pi_{\text{ref}}$:

$$\max_\pi \mathbb{E}_{y \sim \pi(\cdot|x), x \sim \mathcal{D}}\left[r(x, y) - \beta \text{KL}(\pi(\cdot|x)||\pi_{\text{ref}}(\cdot|x))\right], \tag{6.1}$$

---

[4]In our work, we only focused on the regret of one selected action, which slightly differs from the regret in Sekhari et al. (2024).

---

**Algorithm 2** Active Direct Preference Optimization (ADPO)

---

**Require:** Regularization parameter $\beta$, uncertainty threshold $\gamma$, learning rate $\eta$, initial model parameter $\boldsymbol{\theta}_1$, batch size $S$

1: **for** $t = 1, \ldots, T$ **do**
2:   Receive batch of data $\mathcal{B}_t = \{x_i, y_i^1, y_i^2\}_{i=1}^S$
3:   **for** $i = 1, \ldots, S$ **do**
4:     Set the confidence $C_{\boldsymbol{\theta}_t}(x_i, y_i^1, y_i^2)$ as in (6.3)
5:     **if** $C_{\boldsymbol{\theta}_t}(x_i, y_i^1, y_i^2) \leq \gamma$ **then**
6:       Query for the human label and set $o_i$ as the queried preference.
7:     **else**
8:       Set $o_i \leftarrow \text{sign}\big(r_{\boldsymbol{\theta}_t}(x, y^1) - r_{\boldsymbol{\theta}_t}(x, y^2)\big)$
9:     **end if**
10:   **end for**
11:   Update $\boldsymbol{\theta}_{t+1} \leftarrow \boldsymbol{\theta}_t - \eta \nabla_{\boldsymbol{\theta}} \mathcal{L}_{\mathcal{B}_t}(\pi_{\boldsymbol{\theta}_t}, \pi_{\boldsymbol{\theta}_1})$
12: **end for**

---

where $\beta > 0$ is the regularization parameter, $\mathcal{D}$ is the distribution of the prompts and $\pi_{\text{ref}}$ is the reference policy, which corresponds to the SFT checkpoint. The optimal policy of (6.1) is follows:

$$\pi^*(y|x) \propto \pi_{\text{ref}}(y|x) \exp(r(x, y)).$$

Therefore, given the final model parameter $\boldsymbol{\theta}$, we can rewrite the reward in the following form:

$$r_{\boldsymbol{\theta}}(x, y) = \beta\big(\log \pi_{\boldsymbol{\theta}}(y|x) - \log \pi_{\text{ref}}(y|x)\big) + \beta Z(x),$$

where $Z(x)$ is a constant independent of $y$. Plugging $r_{\boldsymbol{\theta}}(x, y)$ into the BT model and fitting the model with the preference labels in dataset $\mathcal{D}$, we get the following DPO training objective:

$$\mathcal{L}_{\text{DPO}}(\pi_{\boldsymbol{\theta}}, \pi_{\text{ref}}) = -\mathbb{E}_{(x, y^1, y^2, o) \sim \mathcal{D}}\Big[\log \sigma\Big(o \cdot \big(r_{\boldsymbol{\theta}}(x, y^1) - r_{\boldsymbol{\theta}}(x, y^2)\big)\Big)\Big],$$

where $y^1$ and $y^2$ are the two responses to the given prompt $x$, and $o$ is the human preference such that $o = 1$ indicates a preference for $y^1$, and $o = -1$ indicates a preference for $y^2$. Compared to standard RLHF, DPO bypasses the reward modeling process and thus eliminates the introduced reward noise.

**Confidence Estimator**   The key to achieving query efficiency in Algorithm 1 is the confidence-based data filter. However, in real applications, rewards are no longer necessarily parameterized by a linear function. Thus, the uncertainty estimator cannot be directly transferred to empirical cases. Since the model is essentially predicting the probability of human preference labels, i.e.,

$$\mathbb{P}(o = 1|x, y^1, y^2) = \sigma(r_{\boldsymbol{\theta}}(x, y^1) - r_{\boldsymbol{\theta}}(x, y^2)), \tag{6.2}$$

where $o$ stands for the preference label and $r_{\boldsymbol{\theta}}$ is the reward model. We can use the reward model's predicted probability as its uncertainty. Specifically, if $|r_{\boldsymbol{\theta}}(x, y^1) - r_{\boldsymbol{\theta}}(x, y^2)|$ is large, then the predicted probability is close to 0 or 1, which means the model is confident about its prediction. Otherwise, if $|r_{\boldsymbol{\theta}}(x, y^1) - r_{\boldsymbol{\theta}}(x, y^2)|$ is close to 0, the predicted probability is close to $1/2$, which indicates the model's confidence is low. Therefore, we define the following function $C_{\boldsymbol{\theta}}$:

$$C_{\boldsymbol{\theta}}(x, y^1, y^2) = |r_{\boldsymbol{\theta}}(x, y^1) - r_{\boldsymbol{\theta}}(x, y^2)|, \tag{6.3}$$

as the confidence level of the model. Such approaches is also applied in previous works like He et al. (2024).

**Training Objectives**   One key design in ADPO is the use of pseudo label, which is inspired by previous methods such as Gentile et al. (2022). For given answer pairs, if the model is very confident in its preference label, we then use the preference label predicted by the model (i.e., pseudo label) for training. To be specific, given a prompt $x$ and the corresponding answers $y^1$ and $y^2$, the predicted preference label can be defined as follows:

$$o_{\boldsymbol{\theta}}(x, y^1, y^2) = \begin{cases} o & \text{if } C_{\boldsymbol{\theta}}(x, y^1, y^2) \leq \gamma \\ \text{sign}\big(r_{\boldsymbol{\theta}}(x, y^1) - r_{\boldsymbol{\theta}}(x, y^2)\big) & \text{if } C_{\boldsymbol{\theta}}(x, y^1, y^2) > \gamma \end{cases}, \tag{6.4}$$

Table 1: Results on objective benchmarks. We use **bold** for the highest score and underline for the second highest. ADPO significantly outperforms DPO on ARC, TruthfulQA, and performs comparably to DPO on HellaSwag, resulting higher average performances. Besides, ADPO only makes 16k queries on Zephyr-Beta and 3.6k queries on Zephyr-Gemma, which is about only a quarter to half of the queries made by DPO.

| Models | ARC | TruthfulQA | HellaSwag | Average | # Queries |
|---|---|---|---|---|---|
| Zephyr-Beta-SFT | 58.28 | 40.36 | 80.72 | 59.79 | 0 |
| Zephyr-Beta-DPO | 61.17 | 45.15 | 82.08 | 62.80 | 62k |
| Zephyr-Beta-APL | 61.35 | 45.15 | 82.17 | 62.89 | 30k |
| Zephyr-Beta-AE-DPO | 61.38 | 45.46 | 82.99 | 63.28 | 28k |
| Zephyr-Beta-ADPO | **62.29** | **52.25** | **83.11** | **65.88** | 16k |
| Zephyr-Gemma-SFT | 55.03 | 46.92 | 81.45 | 61.13 | 0 |
| Zephyr-Gemma-DPO | 58.45 | 52.07 | 83.48 | 64.67 | 6751 |
| Zephyr-Gemma-APL | 59.47 | 52.27 | 83.30 | 65.01 | 3456 |
| Zephyr-Gemma-AE-DPO | 60.41 | 54.03 | **83.68** | 66.04 | 4219 |
| Zephyr-Gemma-ADPO | **61.01** | **57.55** | 83.16 | **67.24** | 3652 |

where $o$ is the human preference upon query, $\text{sign}(z)$ is the signal of $z$ and $\gamma$ is the confidence threshold (corresponding to the threshold $\Gamma$ in APPO, which can be empirically selected by a grid search on a small portion of the dataset). With the predicted preference labels of given prompts and answers, now we can formulate our training objective as the follows:

$$\mathcal{L}_{\mathcal{D}}(\pi_{\boldsymbol{\theta}}, \pi_{\text{ref}}) = -\mathbb{E}_{(x,y^1,y^2)\sim\mathcal{D}}\Big[\log \sigma\Big(o_{\boldsymbol{\theta}}(x,y^1,y^2)\cdot\big(r_{\boldsymbol{\theta}}(x,y^1) - r_{\boldsymbol{\theta}}(x,y^2))\big)\Big)\Big]. \tag{6.5}$$

To make our approach more time efficient in practice, we follow the standard approach in DPO and use mini-batch gradient descent to update the parameters of our model. At each time step, we feed the model with a batch of data $\{(x_i, y_i^1, y_i^2)\}_{i=1}^S$. We then compute the pseudo-labels and update the model parameters by one-step gradient descent.

**Remark 6.1.** Compared to the loss of vanilla DPO, the only additional term in (6.5) is the pseudo-labeling term $o_{\boldsymbol{\theta}}$, which can be directly obtained from the difference of estimated rewards $r_{\boldsymbol{\theta}}(x,y^1) - r_{\boldsymbol{\theta}}(x,y^2)$. Since the difference of estimated rewards is also required in the forward computation of DPO, the computation of $o_{\boldsymbol{\theta}}$ results to almost no extra computational overhead, making ADPO also computationally efficient.

## 7 Experiments

In this section, we conducted extensive experiments to verify the effectiveness of ADPO. Our experiments reveal that ADPO outperforms DPO while requiring only up to half of the queries. Additionally, our ablation studies show that involving pseudo-labels plays a key role in the training process.

### 7.1 Experimental Setup

**Models and Datasets**  We start from two different base models `zephry-7b-sft-full`[5] (Zephyr-Beta-SFT) and `zephyr-7b-gemma-sft-v0.1`[6] (Zephyr-Gemma-SFT), which is supervised-finetuned from Mistral-7B (Jiang et al., 2023) model and gemma-7B (Team et al., 2024) correspondingly. Zephyr-Beta-SFT is obtained by conducting SFT on Ultrachat-200k (Ding et al., 2023) dataset and Zephyr-Gemma-SFT is obtained by conducting SFT on deita-10k-v0-sft (Liu et al., 2023). We follow the approach in alignment-handbook[7] and adopt the corresponding human-preference datasets. In the experiments, we use pre-labeled datasets for simplicity[8]. Specifically, we use Ultrafeedback-binarized (Ding et al., 2023) to train Zephyr-Beta-SFT and dpo-mix-7k[9] to train Zephyr-Gemma-SFT.

---

[5] https://huggingface.co/alignment-handbook/zephyr-7b-sft-full

[6] https://huggingface.co/HuggingFaceH4/zephyr-7b-gemma-sft-v0.1

[7] https://github.com/huggingface/alignment-handbook

[8] In real world scenarios, the label might be queried in-the-loop. While this process does not impose extra cost for automatic labeler (e.g., LLMs), it might be slower than gathering a big dataset of labels all at once for human labelers.

[9] https://huggingface.co/datasets/argilla/dpo-mix-7k

Table 2: Results on subjective benchmarks including AlpacaEval 2.0 and MT-Bench. We use **bold** for the highest score and underline for the second highest. Here WR stands for win rate and LC stands for length controlled. ADPO achieves comparable performance with DPO on starting from Zephyr-Beta-SFT and outperforms DPO starting from Zephyr-Gemma-SFT. The checkpoints evaluated here are exactly the same as those in Table 1, so they share the same number of queries.

| Models | MT-Bench | | | Alpaca Eval 2.0 | | |
|---|---|---|---|---|---|---|
| | First Turn | Second Turn | Average | LC WR | WR | Avg. Length |
| Zephyr-Beta-SFT | 6.82 | 5.94 | 6.39 | 4.59 | 4.69 | 1741 |
| Zephyr-Beta-DPO | **7.55** | **7.27** | **7.41** | **13.57** | **12.67** | 1735 |
| Zephyr-Beta-APL | 6.93 | 6.63 | 6.80 | 10.08 | 9.58 | 1801 |
| Zephyr-Beta-AE-DPO | 6.78 | 6.50 | 6.64 | 11.89 | 11.34 | 1588 |
| Zephyr-Beta-ADPO | 7.31 | 7.08 | 7.20 | 12.67 | 12.02 | 1801 |
| Zephyr-Gemma-SFT | 5.62 | 5.56 | 5.59 | 0.13 | 0.62 | 4296 |
| Zephyr-Gemma-DPO | 5.94 | 5.49 | 5.72 | 3.68 | 10.70 | 9064 |
| Zephyr-Gemma-APL | **6.62** | 5.59 | 6.10 | 2.71 | 11.64 | 17474 |
| Zephyr-Gemma-AE-DPO | 6.31 | 5.59 | 5.96 | 3.54 | **16.22** | 16858 |
| Zephyr-Gemma-ADPO | 6.53 | **6.49** | **6.51** | **3.81** | 15.85 | 8967 |

**Baselines and Evaluation**  We consider DPO as our baseline and use full-finetune to optimize the models for both DPO and ADPO. Please refer to Appendix C for more details regarding the selection of hyperparameters. In addition, we also consider two concurrent similar DPO-based active learning approaches, namely AE-DPO (denoted by AE-DPO) proposed by Mehta et al. (2023) and Active Preference Learning (denoted by APL) proposed by Muldrew et al. (2024), as our baselines. We adopt both objective and subjective evaluation techniques to evaluate the resulting models. Specifically, we employ ARC (Clark et al., 2018), HellaSwag (Zellers et al., 2019) and TruthfulQA (Lin et al., 2022) as benchmarks for objective evaluation. Among these datasets, ARC (Clark et al., 2018) and HellaSwag (Zellers et al., 2019) focus on the language models' capability of commonsense reasoning, while TruthfulQA (Lin et al., 2022) focuses on human falsehood mimic. For subjective benchmarks, we consider AlpacaEval 2.0 (AlpacaEval) and MT-Bench (Zheng et al., 2024). AlpacaEval employs AlpacaFarm (Dubois et al., 2024), which is made up of general human instructions, as its set of prompts. During evaluating, the model responses and the reference response generated by GPT-4-Turbo are fed into a GPT-4-Turbo for preference annotation and the win rate measures the models capability. MT-Bench is composed of 80 high-quality multi-turn open-ended questions covering a variety of topics. The generated answers are also judged by GPT-4, which gives scores directly without comparison. Please refer to Appendix C for more detailed discussion of the datasets and evaluation.

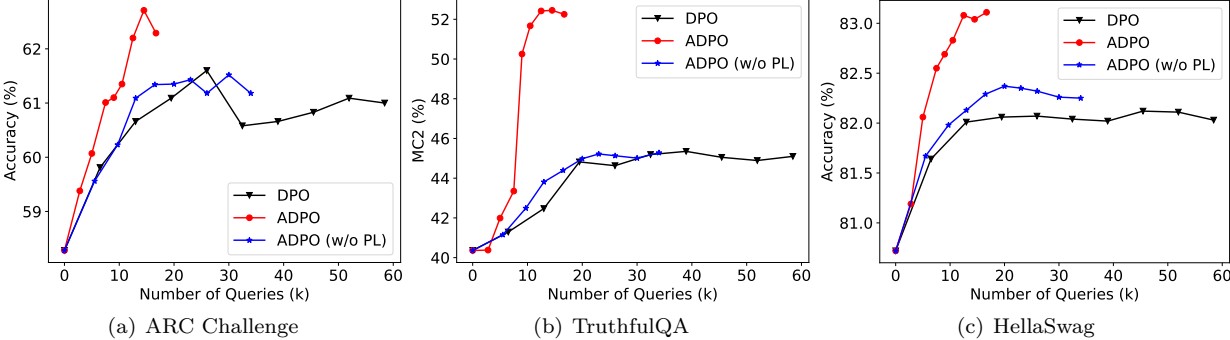

(a) ARC Challenge  (b) TruthfulQA  (c) HellaSwag

Figure 1: The test accuracy curve of DPO and ADPO starting from Zephyr-Beta-SFT. The x-axis is the number of queries and the y-axis is the metric for corresponding dataset. Compared to DPO, ADPO enjoys a faster performance improvement and a higher performance upper bound.

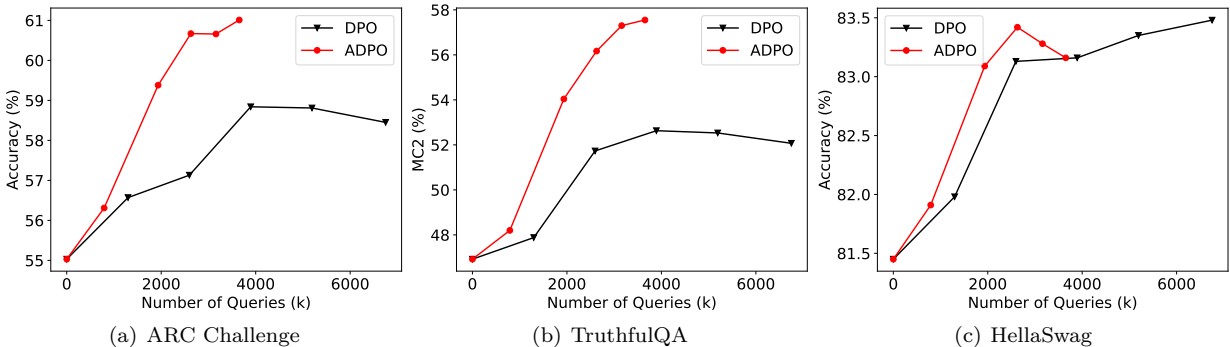

Figure 2: The test accuracy curve of DPO and ADPO starting from Zephyr-Gemma-SFT. The x-axis is the number of queries and the y-axis is the metric for corresponding dataset. Compared to DPO, ADPO enjoys a faster performance improvement and a higher performance upper bound.

## 7.2 Experimental Results

**Results on Objective Benchmarks** The results on subjective benchmarks are presented in Table 1. We see that DPO and ADPO improve the average score by a large margin starting from both Zephyr-Beta-SFT and Zephyr-Gemma-SFT. As for Zephyr-Beta, on TruthfulQA, ADPO outperforms DPO prominently by a margin of 7.1%, and also outperforms DPO on ARC and HellaSwag by 1.08% and 1.03% respectively. Reflecting on the average score, we see that ADPO outperforms DPO by a margin of 3.08%. As for Zephyr-Gemma, ADPO outperforms DPO prominently by a margin of 5.48% on TruthfulQA and 2.56% on ARC. ADPO also reaches a performance comparable to DPO on HellaSwag. Finally, reflecting on the average score, we see that ADPO outperforms DPO by a margin of 2.57%. Furthermore, when comparing to other active-learning-based DPO variants, we see that ADPO outperforms both APL and AE-DPO consistently under both models and datasets except HellaSwag, where the three methods performs almost the same. In terms of average performance, ADPO outperforms the baselines by a significant margin of 2.60% for Zephyr-Beta and 1.2% for Zephyr-Gemma. In summary, results on both models shows the superiority of ADPO. Besides the performance on the benchmarks, we see that ADPO only requires 16k queries for Zephyr-Beta and 3.6k for Zephyr-Gemma, which is only about half of the size of the training dataset and comparable with other active-learning-based DPO methods.

**Results on Subjective Benchmarks** The results on subjective benchmarks are presented in Table 2. For Zephyr-Beta, we see that ADPO achieves comparable performance with DPO. In detail, On MT-Bench, we see that ADPO improves the average performance from 6.39 to 7.20, which is much more significant comparing to its gap with DPO of 0.21. Similarly, on AlpacaEval, ADPO also improve the LC win rate by ar margin of 8.08, which is much more significant than its gap to DPO. As for other baselines, while we see that these methods also significantly improve the performance over the base model, ADPO also outperform these baselines. For Zephyr-Gemma, we see that ADPO outperforms DPO by a considerable margin. In detail, On MT-Bench, we see that ADPO achieves a performance of 6.51 compared 5.72 achieved by DPO. Similarly, on AlpacaEval, ADPO achieves a LC win rate of 15.85, which also surpasses DPO by a large margin. Comparing with other baselines, we see that ADPO outperforms APL and AE-DPO in terms of MT-Bench average score and Alpaca Eval length-controlled win rate.

**Query Efficiency** To further demonstrate the query efficiency for ADPO, we plot the test accuracy curves for ADPO and DPO as the numbers of queries increase on selected datasets. The curves for ARC, HellaSwag and TruthfulQA starting from Zephyr-Beta-SFT and Zephyr-Gemma-SFT are shown in Figure 1 and Figure 2 respectively. For Zephry-Beta, we see that the growth of DPO's performance for ARC almost stops when query number reaches about 30k. This trend can also be observed on TruthfulQA after 20k queries and HellaSwag after about 15k queries. In contrast, the performance of ADPO enjoys a faster improvement when training with the first about 10k results and maintains at a preferable level after that. For Zephyr-Gemma, we observe a similar pattern. The growing speed of the performance of DPO either slows down significantly after making 3k to 4k queries, as shown by Figure 2(a) and Figure 2(b), or maintains at a very low level

(Figure 2(c)). These results suggest that ADPO can effectively select the most informative data and only make queries for these preference labels.

## 8 Ablation Studies

In this section, we consider the impact of the two important parts that are crucial in ADPO, namely pseudo-labeling and the choice of uncertainty threshold. Due to time and computational constraint, all the ablations starts from Zephyr-Beta-SFT and evaluated on objective benchmarks.

### 8.1 Impact of Pseudo Labels

Table 3: The effect of pseudo-labels. ADPO performs better than ADPO (w/o PL) in terms of average scores with fewer queries.

| Model | ARC | TruthfulQA | HellaSwag | Average | # Queries |
|---|---|---|---|---|---|
| DPO | 61.17 | 45.15 | 82.08 | 62.80 | 62k |
| ADPO (w/o PL) | 61.18 | 45.28 | 82.25 | 62.90 | 34k |
| ADPO | **62.29** | **52.25** | **83.11** | **65.88** | **16k** |

An alternative to active learning is to directly follow Algorithm 1 and simply neglect those training data with high confidence. Since neglected samples will not affect the loss and the corresponding gradient, we set the label to 0 so that they will not contribute to $\nabla_{\boldsymbol{\theta}}\mathcal{L}$ in Eq. (6.5) during the learning process. Formally, we define the label $o'_{\boldsymbol{\theta}}$ as follows:

$$o'_{\boldsymbol{\theta}}(x, y^1, y^2) = \begin{cases} o & \text{if } C_{\boldsymbol{\theta}}(x, y^1, y^2) \leq \gamma \\ 0 & \text{if } C_{\boldsymbol{\theta}}(x, y^1, y^2) > \gamma \end{cases}.$$

We keep the remaining part of our method the same and denote this method as "ADPO (w/o PL)". The performance of the trained models is shown in Table 3. We also plot the training curve in Figure 1. The results show that, without pseudo-labels, the performance suffers from a significant downgrade in average score compared to ADPO and does not demonstrate a clear advantage against vanilla DPO. The training curves further indicate that, without pseudo labels, the training dynamics are much more similar to vanilla DPO. These results show that the pseudo-labels plays a crucial role our designed active learning process.

Table 4: The quantitative study of the error rate of the pseudo labels. For each steps, we report the number of all the predicted labels before this step, and the total number of all the correctly predicted labels before this steps. The results show that, as the training proceeds, the correct rate of pseudo-labels remains at a relatively low level below 25%, which demonstrate reliability of the pseudo-labels.

| Training Steps | 50 | 100 | 200 | 300 | 450 | 600 | 750 | 950 |
|---|---|---|---|---|---|---|---|---|
| # Total Predicted Labels | 24 | 952 | 4904 | 9840 | 17448 | 25432 | 33232 | 43904 |
| # Correctly Predicted Labels | 16 | 808 | 3936 | 7792 | 13432 | 19392 | 25384 | 33488 |
| Correct Rate (%) | 66.67 | 84.87 | 80.26 | 79.19 | 76.98 | 76.25 | 76.38 | 76.28 |

We further count the correctly and incorrectly predicted labels as the training proceeds to quantitatively study the quality of the pseudo-labels. The results are compiled in Table 4. The results show that, during the training procedure, about three quarters of all the predicted preference labels align with the ground true. This result shows that when the model has a high confidence, its predictions are generally reliable. Such pseudo labels helps the training process makes use of these samples and therefore enhances data-efficiency.

### 8.2 Value of Confidence Threshold

We study the impact of different confidence thresholds by varying the value of $\gamma$ from 0.2 to 4.0. For each $\gamma$, we count the preference labels used by the models and evaluate the trained models on the objective benchmarks. We plot the results in Figure 3 and also provide some numerical results in Table 5. As shown in Figure 3, when $\gamma$ is small, even about 6k samples are sufficient for ADPO to match the performance of DPO. When $\gamma$ is large, ADPO queries for more labels and the performance become close to DPO. Another

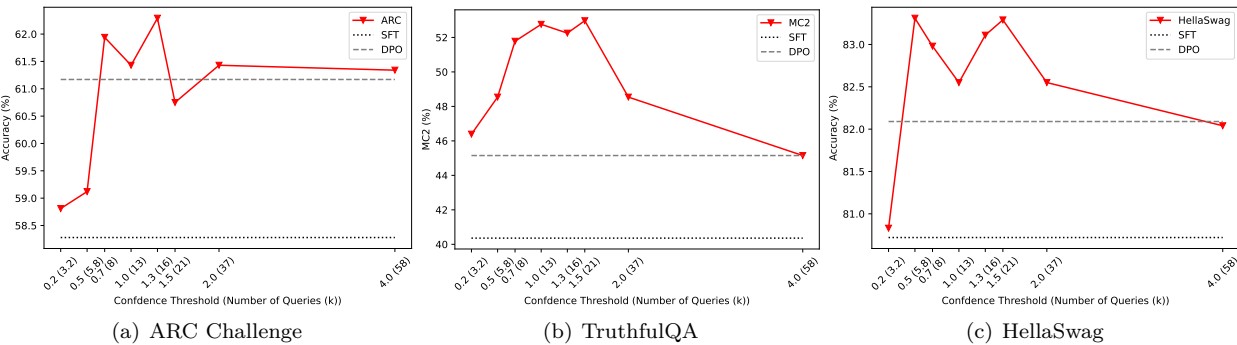

(a) ARC Challenge        (b) TruthfulQA        (c) HellaSwag

Figure 3: The performances on three objective benchmarks and the average performances as $\gamma$ varies from 0.2 to 4.0. The results shows that ADPO performs comparable with DPO over a wide range of $\gamma$ and can matches the performance of DPO even with only fewer than 6k samples. These results demonstrate the robustness of ADPO over $\gamma$.

Table 5: The effect of confidence threshold in ADPO. We vary the value of $\gamma$ and report the evaluation results. When $\gamma$ is increasing, ADPO made more queries and the performance pattern is getting closer to DPO.

| Method | $\gamma$ | ARC | TruthfulQA | HellaSwag | Average | # Queries |
|--------|------|-------|------------|-----------|---------|-----------|
| DPO | - | 61.17 | 45.15 | 82.08 | 62.80 | 62k |
| ADPO | 1.0 | 61.43 | 52.76 | 82.55 | 65.58 | 13k |
| | 1.3 | 62.29 | 52.25 | 83.11 | 65.88 | 16k |
| | 1.5 | 60.75 | 52.97 | 83.29 | 65.67 | 21k |

observation ADPO matches the performance of the DPO baseline over wide range of $\gamma$, which implies that ADPO is not very sensitive to the uncertainty threshold and an coarse grid search of confidence threshold can introduce a fairly good performance.

## 9 Conclusion and Future Work

In this work, we considered query-efficient methods for aligning LLMs with human preference. We first formulated the problem as a contextual dueling bandit. Under linear reward and sub-optimal gap assumption, we proposed an active-learning-based algorithm, APPO. Our theoretical analysis shows that our algorithm enjoys a constant instance-dependent regret upper bound and query complexity. We then adapted our algorithm to direct preference optimization and proposed a query efficient DPO method, ADPO. We conducted experiments starting from two models, Zephyr-Beta-SFT and Zephyr-Gemma-SFT and evaluated the resulting models on both objective benchmarks and subjective benchmarks. Results show that, ADPO achieves a comparable or even better performance compared to DPO with only less than half the demands on the human preference labels. Despite the good performance ADPO achieves, since it uses DPO as the framework of our practical method, our theoretical analysis of APPO cannot be directly applied to ADPO. We leave the theoretical analysis of ADPO as our future work.

**Broader Impact Statement**

This paper studies aligning LLMs with human preference in a query-efficient manner. We believe that this topic has the following social impacts. First, LLM-based chatbots have demonstrated substantial capabilities as AI assistants and they are now increasingly relied upon by individuals. The key step towards building helpful AI assistant is aligning LLMs with human ethics and preferences. Secondly, aligning LLMs requires a large number of human preference labels, necessitating considerable human labor and material resources. In this paper, We propose a query-efficient method to align LLMs with human preference. Our experiments results indicate that our method can better align LLMs with human preference with significantly fewer queries for human preferences. Therefore, we believe that our method potentially alleviate the labor and resource demands within this process.

**Acknowledgments**

We thank the anonymous reviewers and the action editor for their helpful comments. We also thank Qiwei Di for feedback at the early stage of this work. KJ, JH and QG are supported in part by the NSF grants IIS-1906169, DMS-2323113, and IIS-2403400. JH is also supported by UCLA Dissertation Year Fellowship. The views and conclusions contained in this paper are those of the authors and should not be interpreted as representing any funding agencies.

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

## A    Additional Related Work

**Dueling Bandits**   Dueling bandits represent a variant of the multi-armed bandit problem, incorporating preference feedback between two selected arms (Yue et al., 2012). Existing results in this domain generally fall into two categories, shaped by their assumptions about preference probability. The first category of work (Yue et al., 2012; Falahatgar et al., 2017; 2018; Ren et al., 2019; Wu et al., 2022; Lou et al., 2022) assumes a transitivity property for preference probability and focuses on identifying the optimal action. Our work also belongs to this category. The second category of work (Jamieson et al., 2015; Heckel et al., 2018; Saha, 2021; Wu et al., 2024; Dudík et al., 2015; Ramamohan et al., 2016; Balsubramani et al., 2016) focuses on general preferences with various criteria for optimal actions, such as Borda winner and Copeland winner.

Expanding beyond the standard dueling bandit problem, Dudík et al. (2015) was the first to incorporate contextual information into the dueling bandit framework. Subsequently, Saha (2021) studied the $K$-arm

contextual dueling bandit problem and proposed an algorithm with a near-optimal regret guarantee. In order to addressing the challenge of a potentially large action space, Bengs et al. (2022) also considered linear function approximation and extended these results to the contextual linear dueling bandit problem and obtained a regret guarantee of $\widetilde{O}(d\sqrt{T})$. Recently, Di et al. (2024) introduced a layered algorithm, improving the results to a variance-aware guarantee of $\widetilde{O}(d\sqrt{\sum \sigma_t^2})$, where $\sigma_t^2$ denotes the variance of the observed preference in round $t$.

## B  Further Discussions on Related Work

In this section, we provide further discussion on previous works (Wang et al., 2023; Zhan et al., 2023; Wu & Sun, 2023; Sekhari et al., 2024), which consider query complexity in the dueling bandit setting, and explain why they fail to achieve an instance-dependent regret guarantee.

**Comparison with Wang et al. (2023)**  Wang et al. (2023) proposed a general framework, P2R, for efficiently querying human preferences, and later extended it to a white-box algorithm (P-OMLE) with a specialized analysis. However, the P2R algorithm relies on a comparison oracle that is stronger than ours. In the bandit setting, the oracle in Wang et al. (2023) can return preference labels between responses to different prompts, which often exceeds the abilities of typical users. In contrast, our oracle only requires preferences between responses generated from the same prompt. Furthermore, P2R algorithms necessitate multiple independent comparisons between a baseline trajectory and user-generated trajectories, making it impractical to ask a single user for multiple independent preferences on the same query. Our ADPO algorithm, by contrast, only requires one preference feedback per query, making it much more user-friendly. Additionally, in the linear reward setting, the query complexity for the Preference-based OMLE (white-box) algorithm is $\widetilde{O}(d^2/\Delta^2)$[10], which is the same as ours. However, P-OMLE requires solving an optimization problem over complex confidence regions, resulting in an intractable planning phase. In comparison, our APPO algorithm introduces an explicit confidence bonus to bypass this complexity and uses a policy optimization method, which is more tractable and closely aligned with practical RLHF methods, while still achieving the same query complexity.

**Failures in Achieving Instance-Dependent Regret Guarantees**  Recently, Zhan et al. (2023) proposed a pure-exploration style algorithm (REGIME) that can identify the $\epsilon$-optimal policy with a query complexity of $\widetilde{O}(d^2/\epsilon^2)$, where $d$ is the dimension of the feature space. However, it is important to note that the output policy may be a randomized policy, and does not guarantee a constant regret, even under the assumption of a minimal sub-optimality gap $\Delta$. Specifically, the minimal sub-optimality gap does not prevent a randomized policy from incurring regret between 0 and $\Delta$. Thus, even for an $\epsilon$ much smaller than the sub-optimality gap $\Delta$, there is no guarantee that a randomized algorithm will always achieve zero regret. For example, a policy that selects the optimal action with a probability of 50% and a $\Delta$-suboptimal action with a probability of 50% will result in a regret of $\epsilon = \Delta/2$. In this situation, the REGIME algorithm may lead to linear regret with respect to $T$ and fail to achieve instance-dependent regret guarantees.

A similar issue arises when transferring the sample complexity guarantee in Wang et al. (2023) to an instance-dependent regret bound. Additionally, we observe that in Proposition 5 of Wang et al. (2023), the P2R framework achieves finite sample complexity using the UCBVI algorithm (Azar et al., 2017). It is important to note that the original UCBVI algorithm employs deterministic policies for each episode. However, Azar et al. (2017) only provides a $\sqrt{T}$ regret guarantee for the first $T$ rounds, rather than a sample complexity guarantee. To address this, Jin et al. (2018) demonstrates that any algorithm with sublinear regret can derive a finite sample complexity by randomly selecting a policy from the first $T$ rounds, resulting in a final randomized policy. After this step, the output policy may no longer be deterministic and fails to provide instance-dependent regret guarantees.

Another related work, Wu & Sun (2023), proposed a sampling-based algorithm (PR-LSVI) that provides a $\widetilde{O}(d^3\sqrt{T})$ regret guarantee for the first $T$ rounds, which is not directly related to the sub-optimality gap. Consequently, a random mixture over the first $T$ rounds is required to identify a near-optimal policy, and it fails to achieve constant regret even in the presence of a positive sub-optimality gap.

---

[10]The query complexity of P-OMLE has a logarithmic dependency on the reward function space $\mathbb{R}$. For the linear reward function class, where $\log \mathbb{R} = d$, their complexity becomes $d\log \mathbb{R}/\Delta^2 = d^2/\Delta^2$.

As demonstrated above, all of these works can only find a random policy that achieves $\epsilon$-optimality, which cannot provide an instance-dependent regret guarantee, even with the assumption of a minimal sub-optimality gap.

## C  Additional Experiment Details

**Hyper-parameters for Training Zephyr-Beta**   We trained our models on 4×NVIDIA A100 GPUs, with about 80G memory for each GPU. We set the learning rate to 5e-7 for both DPO and ADPO. We use a linear learning rate scheduler with a warm-up ratio of 0.1. The batch size per device is set to 4 and the gradients are accumulated every 4 steps, resulting in equivalent batch size 64. We set dropout to 0.1 and the regularization parameter $\beta = 0.1$ for both DPO and ADPO. For both ADPO and its counterpart without pseudo labels, we set the uncertainty threshold $\gamma = 1.3$. We trained one epoch for both DPO and ADPO, which takes roughly 9 hours for both methods.

**Hyper-parameters for Training Zephyr-Gemma**   We trained our models on 4×NVIDIA A100 GPUs, with about 80G memory for each GPU. We set the learning rate to 5e-7 for both DPO and ADPO. We use a linear learning rate scheduler with a warm-up ratio of 0.1. The batch size per device is set to 4 and the gradients are accumulated every 4 steps, resulting in equivalent batch size 64. We set dropout to 0.1 and the regularization parameter $\beta = 0.1$ for both DPO and ADPO. For ADPO, we set the uncertainty threshold $\gamma = 1.5$. We trained one epoch for both DPO and ADPO, which takes roughly 1 hour for both methods.

**Evaluation Setup**   For subjective evaluation benchmarks, we follow the standard setup specified in the original repositories. For objective benchmarks, we use few-show learning to prompt the LLMs. Specifically, the few-shot number of ARC is set to 25, HellaSwag to 10 and TruthfulQA to 0. We use `acc_norm` as the metric for ARC and HellaSwag, and `mc2` for TruthfulQA.

## D  Additional Experiment Results

In this section, we present the additional results which are obtained by starting from Zephyr-Beta-SFT and optimizing the model with LoRA-finetuning (Hu et al., 2021).

**Experiment Setup**   We trained our models on 4×NVIDIA RTX A6000 GPUs, with about 49G memory for each GPU. We set the LoRA rank to 64, $\alpha = 16$, and dropout to 0.1 and learning rate to 1e-5. We use a linear learning rate scheduler with a warm-up ratio of 0.1. The batch size per device is set to 4 and the gradients are accumulated every 4 steps, resulting in equivalent batch size 64. We set the regularization parameter $\beta = 0.1$ for both DPO and ADPO. For ADPO, we set the uncertainty threshold $\gamma = 1.5$. We trained one epoch for both DPO and ADPO, which takes roughly 7 hours. We only evaluate the obtained checkpoints on objective benchmarks due to the costly nature of calling external large language models as the judge.

Table 6: Result on objective benchmarks for LoRA finetuning on Zephyr-7B-Beta. ADPO significantly outperforms DPO on ARC, TruthfulQA and HellaSwag. Besides, ADPO only makes 32k queries, which is about only a quarter to half of the queries made by DPO.

| Models | ARC | TruthfulQA | HellaSwag | Average | # Queries |
|---|---|---|---|---|---|
| Zephyr-Beta-SFT | 58.28 | 40.36 | 80.72 | 59.79 | 0 |
| Zephyr-Beta-DPO (LoRA) | 60.58 | 41.88 | 82.34 | 61.60 | 62k |
| Zephyr-Beta-ADPO (LoRA) | **61.26** | **45.52** | **83.21** | **63.33** | **32k** |

**Benchmark Performances**   The results are summarized in Table 6. We observe a similar results as for full-finetuning. The results show that both DPO and ADPO improve the average score by a large margin. ADPO outperforms DPO on TruthfulQA by a relatively large margin of 3.64% and also reaches an at-least comparable performance on other three datasets. Finally, reflecting on the average score, we see that ADPO outperforms DPO by a margin of 1.73%. Besides, we see that ADPO only requires 32k queries, which is only about half of the size of the training dataset. These results show that with much less number of queries, ADPO can reach a comparable or even superior performance than DPO, which is consistent with results under full-finetune.

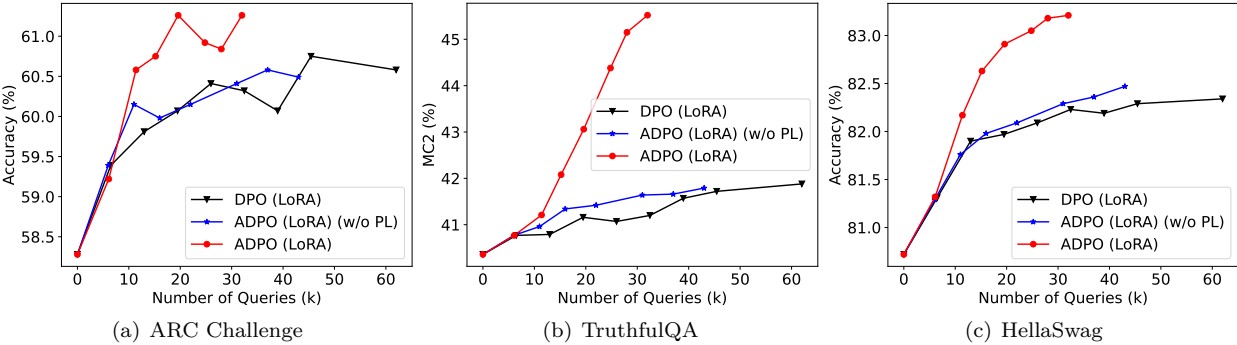

(a) ARC Challenge  (b) TruthfulQA  (c) HellaSwag

Figure 4: The test accuracy curve of DPO, ADPO (w/o PL) and ADPO under LoRA-finetune. The x-axis is the number of queries and the y-axis is the metric for corresponding dataset. Compared to DPO and ADPO (w/o PL), ADPO enjoys a faster performance improvement and a higher performance upper bound.

**Query Efficiency**   We plot a set of similar test accuracy curves for ADPO and DPO for LoRA-finetuning. The results are presented in Figure 4. Here we observe a similar pattern. The growing speed of the performance of DPO either slows down significantly after making 15k to 30k queries, as shown by Figure 4(a) and Figure 4(b), or maintains at a very low level (Figure 4(c)). These results suggest that ADPO can effectively select the most informative data and only make queries for these preference labels.

## D.1   Ablation Studies

Table 7: The effect of pseudo-labels under LoRA-finetune setting. ADPO performs better than ADPO (w/o PL) in terms of average scores with fewer queries.

| Model | $\gamma$ | ARC | TruthfulQA | HellaSwag | Average | # Queries |
|---|---|---|---|---|---|---|
| ADPO (LoRA) (w/o PL) | 0.8 | 60.49 | 41.39 | 82.23 | 61.37 | 38k |
| ADPO (LoRA) (w/o PL) | 1.0 | 60.49 | 41.62 | 82.32 | 61.48 | 40k |
| ADPO (LoRA) (w/o PL) | 1.2 | 60.49 | 41.79 | 82.47 | 61.58 | 43k |
| ADPO (LoRA) | 1.5 | **61.26** | **45.52** | **83.21** | **63.33** | **32k** |

**Impact of Pseudo Labels**   Following the setup in Section 8.1, we consider the counterpart of ADPO without pseudo labels under LoRA finetuning. We pick $\gamma$ to be 0.8, 1.0, and 1.2 for ADPO without pseudo labels. The performances of the trained models are shown in Table 7. We also plot the training curve in Figure 4. The results show that, without pseudo-labels, the performance suffers from a significant downgrade in average score compared to ADPO. The training curves further indicate that, without pseudo labels, the training dynamics are much more similar to vanilla DPO. These results show the crucial role of pseudo-labels in the active learning process.

Table 8: The effect of confidence threshold in practice setting. We vary the value of $\gamma$ and report the evaluation results. When $\gamma$ is increasing, ADPO made more queries and the performance pattern is getting closer to DPO.

| Method | $\gamma$ | ARC | TruthfulQA | HellaSwag | Average | # Queries |
|---|---|---|---|---|---|---|
| DPO (LoRA) | - | 60.58 | 41.88 | 82.34 | 61.60 | 35.7k |
| ADPO (LoRA) | 1.0 | 61.01 | 48.41 | 83.35 | 64.26 | 16k |
| | 1.3 | 61.43 | 47.56 | 83.48 | 64.16 | 24k |
| | 1.5 | 61.26 | 45.52 | 83.21 | 63.33 | 32k |
| | 1.8 | 60.92 | 43.20 | 82.13 | 62.08 | 43k |

**Value of Confidence Threshold**   We also study the impact of different confidence thresholds in LoRA fine-tuning setting. We varies the value of $\gamma$ to 1.0, 1.3, 1.5, and 1.8. For each $\gamma$, we count the preference labels used by the models and evaluate the trained models on the datasets. As shown in Table 8, when the confidence threshold is small, with more predicted labels, these models perform better on the TruthfulQA dataset. On the other hand, when the confidence threshold goes larger, the models are making more queries,

and the performance patterns become closer to the DPO baseline. Another observation is that for all our chosen $\gamma$, ADPO consistently outperforms the DPO baseline, which implies that ADPO is not very sensitive to the uncertainty threshold and an coarse grid search of confidence threshold can introduce a fairly good performance.

# E   Proof of Theorems in Section 5

In this section, we provide the proof of Theorem 5.1 and we first introduce several lemmas. The following lemma provides an upper bound on the query complexity and the corresponding dataset size $|\mathcal{C}_T|$.

**Lemma E.1** (Modified from Lemma 4.5, Zhang et al., 2023). Given a uncertainty threshold $0 < \Gamma \leq 1$, if we set the regularization parameter $\lambda = B^{-2}$, then for each round $t \in [T]$, we have $|\mathcal{C}_t| \leq |\mathcal{C}_T| \leq 16d\Gamma^{-2}\log(3LB\Gamma^{-1})$.

For a finite dataset $\mathcal{C}_T$, the following lemma provides a upper bound for the estimation error between $\widehat{\boldsymbol{\theta}}_t$ and $\boldsymbol{\theta}^*$.

**Lemma E.2.** Suppose we have $\|\boldsymbol{\theta}^*\| \leq B$, $\|\boldsymbol{\phi}(x,y)\| \leq L/2$. Then with probability at least $1 - \delta$, for each round $t \in [T]$, we have

$$\|\boldsymbol{\theta}^* - \widehat{\boldsymbol{\theta}}_t\|_{\Sigma_{t-1}} \leq \frac{1}{\kappa_\mu} \cdot \left(\sqrt{\lambda}B + \sqrt{2d\log(\lambda + |\mathcal{C}_T|L^2/d\lambda\delta)}\right),$$

Based on Lemmas E.1 and E.2, the following auxiliary lemma proposes a proper choice for the uncertainty threshold $\Gamma$ and confidence radius $\beta$ in Algorithm 1.

**Lemma E.3.** If we set the uncertainty threshold $\Gamma = \kappa_\mu\Delta/(2d\iota_1)$ and confidence radius $\beta = \kappa_\mu^{-1}(1 + 4\sqrt{d\iota_2} + \sqrt{2d\iota_3})$, where $\iota_1 = 42\log(126LB\sqrt{d}\Delta^{-1}\kappa_\mu^{-1}) + \sqrt{8\log(1/\delta)}$, $\iota_2 = \log(3LB\Gamma^{-1})$ and $\iota_3 = \log\left((1 + 16L^2B^2\Gamma^{-2}\iota_2)/\delta\right)$, then we have $2\beta\Gamma < \Delta$ and

$$\beta \geq \frac{1}{\kappa_\mu} \cdot \left(\sqrt{\lambda}B + \sqrt{2d\log(\lambda + |\mathcal{C}_T|L^2/d\lambda\delta)}\right).$$

With these parameters, we now define the event $\mathcal{E}_1$ as

$$\mathcal{E}_1 = \{\forall t \in [T], \|\widehat{\boldsymbol{\theta}}_t - \boldsymbol{\theta}^*\|_{\boldsymbol{\Sigma}_{t-1}^{-1}} \leq \beta\}.$$

According to Lemma E.2 and Lemma E.3, we have $\Pr(\mathcal{E}_1) \geq 1 - \delta$. Conditioned on the event $\mathcal{E}_1$, the following lemma suggests that our estimated discrepancy is no less than the actual discrepancy.

**Lemma E.4.** On the event $\mathcal{E}_1$, for each round $t \in [T]$, context $x \in \mathcal{X}$ and any action $y \in \mathcal{A}$, the estimated discrepancy $\widehat{D}_t(x,y)$ satisfied

$$\widehat{D}_t(x,y) \geq D_t(x,y) = \langle \boldsymbol{\theta}^*, \boldsymbol{\phi}(x,y) - \boldsymbol{\phi}_t^2\rangle.$$

On the other hand, we have

$$\widehat{D}_t(x,y) \leq D_t(x,y) + 2\beta\|\boldsymbol{\phi}(x,y) - \boldsymbol{\phi}_t^2\|_{\boldsymbol{\Sigma}_{t-1}^{-1}}.$$

It is worth to notice that in Algorithm 1 (Line 11), we update the policy $\pi_t$ with online mirror descent and the following lemma provides the regret guarantee for this process.

**Lemma E.5** (Modified from Lemma 6.2, He et al., 2022a). For any estimated value function $\widehat{D}_t(x,\cdot)$, if we update the policy $\pi_{t+1}(\cdot|x)$ by the exponential rule:

$$\pi_{t+1}(\cdot|x) \propto \pi_t(\cdot|x) \cdot \exp\left(\eta\widehat{D}_t(x,\cdot)\right), \tag{E.1}$$

then the expected sub-optimality gap at round $T$ can be upper bounded as follows:

$$\mathbb{E}_{x\sim\mathcal{D},y\sim\pi^*(\cdot|x)}[\widehat{D}_t(x,y)] - \mathbb{E}_{x\sim\mathcal{D},y\sim\pi_t(\cdot|x)}[\widehat{D}_t(x,y)]$$
$$\leq 2\eta + \eta^{-1}\mathbb{E}_{x\sim\mathcal{D}}\left[\mathrm{KL}\left(\pi^*(\cdot|x)\|\pi_t(\cdot|x)\right) - \mathrm{KL}\left(\pi^*(\cdot|x)\|\pi_{t+1}(\cdot|x)\right)\right]$$

With the help of these lemmas, we are now ready to prove our main theorem.

*Proof of Theorem 5.1.* Now we start the regret analysis. For simplicity, for each round $t \in [T]$, we use $\phi_t$ to denote $\phi(x_t, y_t)$. Initially, the episodes and their corresponding regret can be decomposed into two groups based on whether episode $t$ is added to the dataset $\mathcal{C}_T$:

$$
\begin{aligned}
\text{Regret}(T) &= \sum_{t=1}^{T} \langle \boldsymbol{\theta}^*, \phi_t^* \rangle - \langle \boldsymbol{\theta}^*, \phi_t^1 \rangle \\
&= \sum_{t=1}^{T} D_t(x_t, y_t^*) - D_t(x_t, y_t^1) \\
&= \underbrace{\sum_{t \in \mathcal{C}_T} D_t(x_t, y_t^*) - D_t(x_t, y_t^1)}_{I_1} + \underbrace{\sum_{t \notin \mathcal{C}_T} D_t(x_t, y_t^*) - D_t(x_t, y_t^1)}_{I_2}
\end{aligned}
\tag{E.2}
$$

where $D_t(x, y) = \langle \boldsymbol{\theta}^*, \phi(x, y) - \phi_t^2 \rangle$ denotes the reward gap between action $y \in \mathcal{A}$ and selected action $y_t^2$ at round $t$.

Now, we bound this two term separately. For the term $I_1$, we have

$$
\begin{aligned}
I_1 &= \sum_{t \in \mathcal{C}_T} D_t(x_t, y_t^*) - D_t(x_t, y_t^1) \\
&\leq \underbrace{\sum_{t \in \mathcal{C}_T} \widehat{D}_t(x_t, y_t^1) - D_t(x_t, y_t^1)}_{J_1} + \sum_{t \in \mathcal{C}_T} \widehat{D}_t(x_t, y_t^*) - \widehat{D}_t(x_t, y_t^1) \\
&= J_1 + \underbrace{\sum_{t \in \mathcal{C}_T} \mathbb{E}_{x_t \sim \mathcal{D}, y \sim \pi^*(\cdot|x)}[\widehat{D}_t(x_t, y)] - \mathbb{E}_{x_t \sim \mathcal{D}, y \sim \pi_t(\cdot|x)}[\widehat{D}_t(x_t, y)]}_{J_2} \\
&\quad + \underbrace{\sum_{t \in \mathcal{C}_T} \widehat{D}_t(x_t, y_t^*) - \widehat{D}_t(x_t, y_t^1) - \sum_{t \in \mathcal{C}_T} \mathbb{E}_{x_t \sim \mathcal{D}, y \sim \pi^*(\cdot|x)}[\widehat{D}_t(x_t, y)] - \mathbb{E}_{x_t \sim \mathcal{D}, y \sim \pi_t(\cdot|x)}[\widehat{D}_t(x_t, y_t)]}_{J_3}, \quad \text{(E.3)}
\end{aligned}
$$

where the inequality holds due to Lemma E.4.

For the term $J_1$, we have

$$
\begin{aligned}
J_1 &= \sum_{t \in \mathcal{C}_T} \widehat{D}_t(x_t, y_t^1) - D_t(x_t, y_t^1) \\
&\leq \sum_{t \in \mathcal{C}_T} \min\{4, 2\beta \|\phi_t^1 - \phi_t^2\|_{\boldsymbol{\Sigma}_{t-1}^{-1}}\} \\
&\leq 4\beta \sqrt{|\mathcal{C}_T| \cdot \sum_{t \in \mathcal{C}_T} \min\{1, \|\phi_t^1 - \phi_t^2\|_{\boldsymbol{\Sigma}_{t-1}^{-1}}^2\}} \\
&\leq 8\beta \sqrt{|\mathcal{C}_T| d \log\left(\frac{\lambda d + |\mathcal{C}_T| L^2}{\lambda d}\right)},
\end{aligned}
\tag{E.4}
$$

where the first inequality holds due to Lemma E.4 with the fact that $-2 \leq D_t(x_y, y_t^1) \leq 2$, the second inequality holds due to Cauchy–Schwarz inequality and the last inequality holds due to the elliptical potential lemma (Lemma G.7).

The term $J_2$ reflects the sub-optimality from the online mirror descent process and can be upper bounded by Lemma E.5. For simplicity, we denote $\mathcal{C}_T = \{t_1, ..., t_K\}$ where $K = |\mathcal{C}_T|$. Thus, we have

$$
\begin{aligned}
J_2 &= \sum_{k=1}^{K} \mathbb{E}_{x_{t_k} \sim \mathcal{D}, y \sim \pi^*(\cdot|x)}[\widehat{D}_{t_k}(x_{t_k}, y)] - \mathbb{E}_{x_{t_k} \sim \mathcal{D}, y \sim \pi_{t_k}(\cdot|x)}[\widehat{D}_{t_k}(x_{t_k}, y)] \\
&\leq \sum_{k=1}^{K} \left( 2\eta + \eta^{-1} \mathbb{E}_{x \sim \mathcal{D}}\left[ \mathrm{KL}(\pi^*(\cdot|x)\|\pi_{t_k}(\cdot|x)) - \mathrm{KL}(\pi^*(\cdot|x)\|\pi_{t_k+1}(\cdot|x)) \right] \right) \\
&= 2\eta K + \eta^{-1} \mathbb{E}_{x \sim \mathcal{D}}\left[ \mathrm{KL}(\pi^*(\cdot|x)\|\pi_1(\cdot|x)) - \mathrm{KL}(\pi^*(\cdot|x)\|\pi_{t_K+1}(\cdot|x)) \right] \\
&\leq 2\eta K + \eta^{-1} \mathbb{E}_{x \sim \mathcal{D}}\left[ \mathrm{KL}(\pi^*(\cdot|x)\|\pi_1(\cdot|x)) \right] \\
&\leq 2\sqrt{32 d \Gamma^{-2} \log(3LB\Gamma^{-1}) \log|\mathcal{A}|}, 
\end{aligned}
\tag{E.5}
$$

where the first inequality holds due to Lemma E.5, the second equation holds due to policy $\pi$ keeps unchanged for $t \in \mathcal{C}_T$, the second inequality holds due to $\mathrm{KL}(\cdot\|\cdot) \geq 0$ and the last inequality holds due to $\eta = \sqrt{\Gamma^2 \log \mathcal{A} / (32 d \log(3LB\Gamma^{-1}))}$ with the fact that $\pi_1$ is uniform policy.

According to Azuma-Hoeffding's inequality (Lemma G.6), with probability at least $1 - \delta$, the term $J_3$ can be upper bounded by

$$
J_3 \leq 2\sqrt{2|\mathcal{C}_T| \log(1/\delta)}.
\tag{E.6}
$$

Substituting (E.4), (E.5) and (E.6) into (E.3), we have

$$
\begin{aligned}
I_1 = J_1 + J_2 + J_3 &\leq 8\beta \sqrt{|\mathcal{C}_T| d \log\left( \frac{\lambda d + |\mathcal{C}_T| L^2}{\lambda d} \right)} + 2\sqrt{32 d \Gamma^{-2} \log(3LB\Gamma^{-1}) \log|\mathcal{A}|} + 2\sqrt{2|\mathcal{C}_T| \log(1/\delta)} \\
&\leq \widetilde{O}(\beta d / \Gamma) \\
&= \widetilde{O}\left( \frac{d^2}{\Delta} \right).
\end{aligned}
\tag{E.7}
$$

where the last inequality holds due to Lemma E.1.

Now, we only need to focus on the term $I_2$. For each round $t \notin \mathcal{C}_T$, we have

$$
\begin{aligned}
D_t(x_t, y_t^*) - D_t(x_t, y_t^1) &= \langle \boldsymbol{\theta}^* - \widehat{\boldsymbol{\theta}}_t, \boldsymbol{\phi}_t^* - \boldsymbol{\phi}_t^2 \rangle + \langle \widehat{\boldsymbol{\theta}}_t, \boldsymbol{\phi}_t^* - \boldsymbol{\phi}_t^2 \rangle - \langle \boldsymbol{\theta}^*, \boldsymbol{\phi}_t^1 - \boldsymbol{\phi}_t^2 \rangle \\
&\leq \beta \|\boldsymbol{\phi}_t^* - \boldsymbol{\phi}_t^2\|_{\boldsymbol{\Sigma}_{t-1}^{-1}} + \langle \widehat{\boldsymbol{\theta}}_t, \boldsymbol{\phi}_t^* - \boldsymbol{\phi}_t^2 \rangle - \langle \boldsymbol{\theta}^*, \boldsymbol{\phi}_t^1 - \boldsymbol{\phi}_t^2 \rangle \\
&\leq \beta \|\boldsymbol{\phi}_t^1 - \boldsymbol{\phi}_t^2\|_{\boldsymbol{\Sigma}_{t-1}^{-1}} + \langle \widehat{\boldsymbol{\theta}}_t, \boldsymbol{\phi}_t^1 - \boldsymbol{\phi}_t^2 \rangle - \langle \boldsymbol{\theta}^*, \boldsymbol{\phi}_t^1 - \boldsymbol{\phi}_t^2 \rangle \\
&\leq 2\beta \|\boldsymbol{\phi}_t^1 - \boldsymbol{\phi}_t^2\|_{\boldsymbol{\Sigma}_{t-1}^{-1}},
\end{aligned}
$$

where the first inequality holds due to Lemma E.4, the second inequality holds due to the selection rule of action $\boldsymbol{\phi}_t^1$ and the last inequality holds due to Lemma E.4. According to the definition of set $\mathcal{C}_T$ in Algorithm 1, for each round $t \notin \mathcal{C}_T$, we have $\|\boldsymbol{\phi}_t^1 - \boldsymbol{\phi}_t\|_{\boldsymbol{\Sigma}_{t-1}^{-1}} \leq \Gamma$. Therefore, the sub-optimality gap at round $t$ is upper bounded by

$$
2\beta \|\boldsymbol{\phi}_t^1 - \boldsymbol{\phi}_t\|_{\boldsymbol{\Sigma}_{t-1}^{-1}} \leq 2\beta\Gamma < \Delta,
$$

where the second inequality holds due to Lemma E.3. According to the minimal sub-optimality assumption (Assumption 3.4), this indicates that the regret yielded in round $t \notin \mathcal{C}_T$ is 0. Summing up over $t \notin \mathcal{C}_T$, we have

$$
I_2 = \sum_{t \in \mathcal{T}_t} D_t(x_t, y_t^*) - D_t(x_t, y_t^1) = 0.
\tag{E.8}
$$

Combining the results in (E.7) and (E.8), we complete the proof of Theorem 5.1. □

# F    Proof of Lemmas in Appendix E

In this section, we provide the proofs of the lemmas in Appendix E.

## F.1    Proof of Lemma E.1

*Proof of Lemma E.1.* The proof follows the proof in Zhang et al. (2023). Here we fix the round $t$ to be $T$ in the proof and only provide the upper bound of $\mathcal{C}_T$ due to the fact that $\mathcal{C}_t$ is monotonically increasing w.r.t. the round $t$. For all selected episode $t \in \mathcal{C}_T$, since we have $\|\phi_t^1 - \phi_t^2\|_{\Sigma_{t-1}^{-1}} \geq \Gamma$, the summation of the bonuses over all the selected episode $t \in \mathcal{C}_T$ is lower bounded by

$$\sum_{t \in \mathcal{C}_T} \min\left\{1, \|\phi_t^1 - \phi_t^2\|_{\Sigma_{t-1}^{-1}}^2\right\} \geq |\mathcal{C}_T| \min\{1, \Gamma^2\} = |\mathcal{C}_T|\Gamma^2, \tag{F.1}$$

where the last equation holds due to $0 \leq \Gamma \leq 1$. On the other hand, according to Lemma G.3, the summation is upper bounded by:

$$\sum_{t \in \mathcal{C}_T} \min\left\{1, \|\phi_t^1 - \phi_t^2\|_{\Sigma_{t-1}^{-1}}^2\right\} \leq 2d \log\left(\frac{\lambda d + |\mathcal{C}_T| L^2}{\lambda d}\right). \tag{F.2}$$

Combining (F.1) and (F.2), we know that the total number of the selected data points $|\mathcal{C}_T|$ satisfies the following inequality:

$$\Gamma^2 |\mathcal{C}_T| \leq 2d \log\left(\frac{\lambda d + |\mathcal{C}_T| L^2}{\lambda d}\right).$$

For simplicity, we reorganized the result as follows:

$$\frac{\Gamma^2 |\mathcal{C}_T|}{2d} \leq \log\left(1 + \frac{2L^2}{\Gamma^2 \lambda} \frac{\Gamma^2 |\mathcal{C}_T|}{2d}\right). \tag{F.3}$$

Notice that $\lambda = B^{-2}$ and $2L^2 B^2 \geq 2 \geq \Gamma^2$, therefore, if $|\mathcal{C}_T|$ is too large such that

$$\frac{\Gamma^2 |\mathcal{C}_T|}{2d} > 4 \log\left(\frac{4L^2 B^2}{\Gamma^2}\right) + 1 \geq 4 \log\left(\frac{4L^2 B^2}{\Gamma^2}\right) + \frac{\Gamma^2}{2L^2 B^2},$$

then according to Lemma G.1, (F.3) will not hold. Thus the necessary condition for (F.3) to hold is:

$$\frac{\Gamma^2 |\mathcal{C}_T|}{2d} \leq 4 \log\left(\frac{4L^2 B^2}{\Gamma^2}\right) + 1 = 8 \log\left(\frac{2LB}{\Gamma}\right) + \log(e) = 8 \log\left(\frac{2LBe^{\frac{1}{8}}}{\Gamma}\right) < 8 \log\left(\frac{3LB}{\Gamma}\right).$$

Applying basic calculus, we obtain the claimed bound for $|\mathcal{C}_T|$ and thus complete the proof of Lemma E.1. □

## F.2    Proof of Lemma E.2

*Proof of Lemma E.2.* This proof follows the proof in Di et al. (2024). For each round $t \in [T]$, we define the following auxiliary quantities:

$$G_t(\boldsymbol{\theta}) = \lambda \boldsymbol{\theta} + \sum_{\tau \in \mathcal{C}_{t-1}} \left[\mu\left((\phi_\tau^1 - \phi_\tau^2)^\top \boldsymbol{\theta}\right) - \mu\left((\phi_\tau^1 - \phi_\tau^2)^\top \boldsymbol{\theta}^*\right)\right](\phi_\tau^1 - \phi_\tau^2)$$

$$\epsilon_t = o_t - \mu\left((\phi_t^1 - \phi_t^2)^\top \boldsymbol{\theta}^*\right)$$

$$Z_t = \sum_{\tau \in \mathcal{C}_{t-1}} \epsilon_\tau (\phi_\tau^1 - \phi_\tau^2).$$

By defining $\widehat{\boldsymbol{\theta}}_t$ as the solution to (4.1), we plug the equation into the definition of $G_t$ and we have

$$G_t(\widehat{\boldsymbol{\theta}}_t) = \lambda \widehat{\boldsymbol{\theta}}_t + \sum_{\tau \in \mathcal{C}_{t-1}} \left[\mu\left((\phi_\tau^1 - \phi_\tau^2)^\top \widehat{\boldsymbol{\theta}}_t\right) - o_\tau + o_\tau - \mu\left((\phi_\tau^1 - \phi_\tau^2)^\top \boldsymbol{\theta}^*\right)\right](\phi_\tau^1 - \phi_\tau^2)$$

$$= \lambda \widehat{\boldsymbol{\theta}}_t + \sum_{\tau \in \mathcal{C}_{t-1}} \left[\mu\left((\phi_\tau^1 - \phi_\tau^2)^\top \widehat{\boldsymbol{\theta}}_t\right) - o_\tau\right](\phi_\tau^1 - \phi_\tau^2) + \sum_{\tau \in \mathcal{C}_{t-1}} \left[o_\tau - \mu\left((\phi_\tau^1 - \phi_\tau^2)^\top \boldsymbol{\theta}^*\right)\right](\phi_\tau^1 - \phi_\tau^2)$$

$$= Z_t.$$

Therefore, we have that

$$G_t(\widehat{\boldsymbol{\theta}}_t) - G_t(\boldsymbol{\theta}^*) = Z_t - G_t(\boldsymbol{\theta}^*) = Z_t - \lambda\boldsymbol{\theta}^*.$$

On the other hand, applying Taylor's expansion, there exists $\alpha \in [0,1]$ and $\widetilde{\boldsymbol{\theta}}_t = \alpha\widehat{\boldsymbol{\theta}}_t + (1-\alpha)\boldsymbol{\theta}^*$, such that the following equation holds:

$$G_t(\widehat{\boldsymbol{\theta}}_t) - G_t(\boldsymbol{\theta}^*) = \lambda(\widehat{\boldsymbol{\theta}}_t - \boldsymbol{\theta}^*) + \sum_{\tau \in \mathcal{C}_{t-1}} \left[ \mu\big((\boldsymbol{\phi}_\tau^1 - \boldsymbol{\phi}_\tau^2)^\top\boldsymbol{\theta}\big) - \mu\big((\boldsymbol{\phi}_\tau^1 - \boldsymbol{\phi}_\tau^2)^\top\boldsymbol{\theta}^*\big) \right](\boldsymbol{\phi}_\tau^1 - \boldsymbol{\phi}_\tau^2)$$

$$= \left[ \lambda\mathbf{I} + \sum_{\tau \in \mathcal{C}_{t-1}} \mu'\big((\boldsymbol{\phi}_\tau^1 - \boldsymbol{\phi}_\tau^2)^\top\widetilde{\boldsymbol{\theta}}_t\big)(\boldsymbol{\phi}_\tau^1 - \boldsymbol{\phi}_\tau^2)(\boldsymbol{\phi}_\tau^1 - \boldsymbol{\phi}_\tau^2)^\top \right](\widehat{\boldsymbol{\theta}}_t - \boldsymbol{\theta}^*)$$

$$= F(\widetilde{\boldsymbol{\theta}}_t)(\widehat{\boldsymbol{\theta}}_t - \boldsymbol{\theta}^*),$$

where we define $F(\widetilde{\boldsymbol{\theta}}_t) = \lambda\mathbf{I} + \sum_{\tau \in \mathcal{C}_{t-1}} \mu'\big((\boldsymbol{\phi}_\tau^1 - \boldsymbol{\phi}_\tau^2)^\top\widetilde{\boldsymbol{\theta}}_t\big)(\boldsymbol{\phi}_\tau^1 - \boldsymbol{\phi}_\tau^2)(\boldsymbol{\phi}_\tau^1 - \boldsymbol{\phi}_\tau^2)^\top$. Thus, we have:

$$\|\widehat{\boldsymbol{\theta}}_t - \boldsymbol{\theta}^*\|_{\boldsymbol{\Sigma}_{t-1}}^2 = (Z_t - \lambda\boldsymbol{\theta}^*)^\top F(\widetilde{\boldsymbol{\theta}}_t)^{-1}\boldsymbol{\Sigma}_{t-1}F(\widetilde{\boldsymbol{\theta}}_t)^{-1}(Z_t - \lambda\boldsymbol{\theta}^*)$$

$$\leq \frac{1}{\kappa_\mu^2}(Z_t - \lambda\boldsymbol{\theta}^*)^\top\boldsymbol{\Sigma}_{t-1}^{-1}(Z_t - \lambda\boldsymbol{\theta}^*)$$

$$= \frac{1}{\kappa_\mu^2}\|Z_t - \lambda\boldsymbol{\theta}^*\|_{\boldsymbol{\Sigma}_{t-1}^{-1}}^2$$

where the inequality holds due to $F(\widetilde{\boldsymbol{\theta}}_t) \succeq \kappa_\mu\widehat{\boldsymbol{\Sigma}}_{t-1}$. Now we have:

$$\|\widehat{\boldsymbol{\theta}}_t - \boldsymbol{\theta}^*\|_{\boldsymbol{\Sigma}_{t-1}} \leq \frac{1}{\kappa_\mu}\|Z_t - \lambda\boldsymbol{\theta}^*\|_{\boldsymbol{\Sigma}_{t-1}^{-1}}$$

$$\leq \frac{1}{\kappa_\mu}\big(\|Z_t\|_{\boldsymbol{\Sigma}_{t-1}^{-1}} + \|\lambda\boldsymbol{\theta}^*\|_{\boldsymbol{\Sigma}_{t-1}^{-1}}\big)$$

$$\leq \frac{1}{\kappa_\mu}\big(\|Z_t\|_{\boldsymbol{\Sigma}_{t-1}^{-1}} + \sqrt{\lambda}B\big),$$

where the second inequality holds due to triangle inequality and last inequality holds due to $\boldsymbol{\Sigma}_{t-1} \succeq \lambda\mathbf{I}$. Now we only need to bound $\|Z_t\|_{\boldsymbol{\Sigma}_{t-1}^{-1}}$.

According to Lemma G.2, with probability at least $1 - \delta$, we have

$$\|Z_t\|_{\boldsymbol{\Sigma}_{t-1}^{-1}} \leq \sqrt{2\log\left(\frac{\sqrt{\det(\boldsymbol{\Sigma}_{t-1})}}{\sqrt{\det(\boldsymbol{\Sigma}_0)}\delta}\right)} \leq \sqrt{2\log\left(\frac{\det(\boldsymbol{\Sigma}_{t-1})}{\lambda^d\delta}\right)} \leq \sqrt{2d\log\left(\frac{\lambda + |\mathcal{C}_t|L^2/d}{\lambda^d\delta}\right)}$$

where the first inequality holds due to Lemma G.2 and the last inequality holds due to Lemma G.5. Now we combine the two term and have

$$\|\widehat{\boldsymbol{\theta}}_t - \boldsymbol{\theta}^*\|_{\boldsymbol{\Sigma}_{t-1}} \leq \frac{1}{\kappa_\mu}\left(\sqrt{\lambda}B + \sqrt{2d\log\left(\frac{\lambda + |\mathcal{C}_t|L^2/d}{\lambda^d\delta}\right)}\right),$$

which concludes our statement. $\qquad\square$

### F.3 Proof of Lemma E.3

*Proof of Lemma E.3.* This proof follows the proof in Zhang et al. (2023). First, we recall that $\Gamma = \Delta\kappa_\mu/2d\iota_1$ and $\beta = \kappa_\mu^{-1}(1 + 4\sqrt{d\iota_2} + \sqrt{2d\iota_3})$. We will first demonstrate that the selection of $\beta$ satisfy the requirement in Lemma E.3. Recalling that $\lambda = B^{-2}$, through basic calculation, we have

$$\kappa_\mu\beta \geq 1 + \sqrt{2d\log\big((1 + L^2B^216d\Gamma^{-2}\iota_2)/d\delta\big)}$$

$$\geq 1 + \sqrt{2d\log\big((1 + L^2B^2|\mathcal{C}_T|)/d\delta\big)}$$

$$= \sqrt{\lambda}B + \sqrt{2d\log(\lambda + |\mathcal{C}_T|L^2/d\lambda\delta)},$$

where the first inequality holds by neglecting the positive term $4\sqrt{d}\iota_2$ and $d \geq 1$, the second inequality holds due to Lemma E.1 and the last equation holds by plugging in $\lambda = B^{-2}$. Now we come to the second statement. First, by basic computation, we have

$$\sqrt{2\iota_3} \leq \sqrt{2\log((1 + 16L^2B^2\Gamma^{-2}\iota_2)} + \sqrt{2\log(1/\delta))}.$$

Notice that we have $L \geq 1$, $B \geq 1$, and $\Gamma \leq 1$, which further implies that $LB\Gamma^{-1} \geq 1$, leading to

$$2 + 4\sqrt{\iota_2} \leq 6\iota_2, \quad \sqrt{2\log((1 + 16L^2B^2\Gamma^{-2}\iota_2)} \leq 3\iota_2.$$

Therefore, we have:

$$2 + 4\sqrt{\iota_2} + \sqrt{2\iota_3} \leq 9\iota_2 + 2\sqrt{\log(1/\delta)}$$
$$\leq 9\log(6LB\sqrt{d}\Delta^{-1}\kappa_\mu^{-1}\iota_1) + 2\sqrt{\log(1/\delta)}.$$

By Lemma G.1, we can identify the sufficient condition for the following inequality

$$(6LB\sqrt{d}\Delta^{-1}\kappa_\mu^{-1})\iota_1 \geq 9(6LB\sqrt{d}\Delta^{-1}\kappa_\mu^{-1})\log(6LB\sqrt{d}\Delta^{-1}\kappa_\mu^{-1}\iota_1) + 2(6LB\sqrt{d}\Delta^{-1}\kappa_\mu^{-1})\sqrt{\log(1/\delta)} \quad \text{(F.4)}$$

is that

$$\iota_1 \geq 36\log(108LB\sqrt{d}\Delta^{-1}\kappa_{mu}^{-1}) + \sqrt{8\log(1/\delta)},$$

which naturally holds due to our definition of $\iota_1$. Eliminating the $6LB\sqrt{d}\Delta^{-1}\kappa_\mu^{-1}$ term in (F.4) yields that

$$\iota_1 \geq 2 + 4\sqrt{\iota_2} + \sqrt{2\iota_3},$$

which implies that

$$2\beta\Gamma = \frac{\Delta\kappa_\mu}{2\sqrt{d}\iota_1}\frac{1}{\kappa_\mu}(1 + 2\sqrt{d}\iota_2 + \sqrt{2\iota_3}) < \Delta.$$

Thus, we complete the proof of Lemma E.3. □

## F.4 Proof of Lemma E.4

*Proof of Lemma E.4.* For each context $x \in \mathcal{X}$ and action $y \in \mathcal{A}$, we have

$$|D_t(x,y) - \langle\widehat{\boldsymbol{\theta}}_t, \boldsymbol{\phi}(x,y) - \boldsymbol{\phi}_t^2\rangle| = |\langle\widehat{\boldsymbol{\theta}}_t - \boldsymbol{\theta}^*, \boldsymbol{\phi}(x,y) - \boldsymbol{\phi}_t^2\rangle|$$
$$\leq \|\widehat{\boldsymbol{\theta}}_t - \boldsymbol{\theta}^*\|_{\boldsymbol{\Sigma}_{t-1}^{-1}} \cdot \|\boldsymbol{\phi}(x,y) - \boldsymbol{\phi}_t^2\|_{\boldsymbol{\Sigma}_{t-1}}$$
$$\leq \beta\|\boldsymbol{\phi}(x,y) - \boldsymbol{\phi}_t^2\|_{\boldsymbol{\Sigma}_{t-1}}, \quad \text{(F.5)}$$

where the first inequality holds due to Cauchy–Schwarz inequality and the second inequality holds due to event $\mathcal{E}_1$. Therefore, we have

$$\langle\widehat{\boldsymbol{\theta}}_t, \boldsymbol{\phi}(x,y) - \boldsymbol{\phi}_t^2\rangle + \beta\|\boldsymbol{\phi}(x,y) - \boldsymbol{\phi}_t^2\|_{\boldsymbol{\Sigma}_{t-1}} \geq D_t(x,y),$$

where the first inequality holds due to (F.5). In addition, we have $D_t(x,y) = \langle\boldsymbol{\theta}^*, \boldsymbol{\phi}(x,y) - \boldsymbol{\phi}_t^2\rangle \leq 2$. Combing these two results, we have

$$\widehat{D}_t(x,y) = \min\{\langle\widehat{\boldsymbol{\theta}}_t, \boldsymbol{\phi}(x,y) - \boldsymbol{\phi}_t^2\rangle + \beta\|\boldsymbol{\phi}(x,y) - \boldsymbol{\phi}_t^2\|_{\boldsymbol{\Sigma}_{t-1}}, 2\} \geq D_t(x,y).$$

On the other hand, we have

$$\widehat{D}_t(x,y) \leq \langle\widehat{\boldsymbol{\theta}}_t, \boldsymbol{\phi}(x,y) - \boldsymbol{\phi}_t^2\rangle + \beta\|\boldsymbol{\phi}(x,y) - \boldsymbol{\phi}_t^2\|_{\boldsymbol{\Sigma}_{t-1}} \leq D_t(x,y) + 2\beta\|\boldsymbol{\phi}(x,y) - \boldsymbol{\phi}_t^2\|_{\boldsymbol{\Sigma}_{t-1}},$$

where the first inequality holds due to the definition of $\widehat{D}_t(x,y)$ and the second inequality holds due to (F.5). Thus, we complete the proof of Lemma E.4. □

## F.5 Proof of Lemma E.5

*Proof of Lemma E.5.* The proof follows the approach in He et al. (2022a). Recall that we assume the policy is updated in round $t$ according to the update rule (E.1), for all contexts $x \in \mathcal{X}$. Thus, we have:

$$\exp\left\{\eta \widehat{D}_t(x,y)\right\} = \frac{\pi_t(y|x)\exp\left\{\eta \widehat{D}_t(x,y)\right\}}{\pi_t(y|x)} = \frac{\rho \pi_{t+1}(y|x)}{\pi_t(y|x)}, \tag{F.6}$$

where $\rho = \sum_{y\in\mathcal{A}} \pi_t(y|x)\exp\left\{\eta \widehat{D}_t(x,y)\right\}$ is the regularization term that is the same for all actions $y \in \mathcal{A}$. Therefore, we have

$$\begin{aligned}
&\sum_{y\in\mathcal{A}} \eta \widehat{D}_t(x,y)\big(\pi^*(y|x) - \pi_{t+1}(y|x)\big) \\
&= \sum_{y\in\mathcal{A}} \big(\log\rho + \log\pi_{t+1}(y|x) - \log\pi_t(y|x)\big)\big(\pi^*(y|x) - \pi_{t+1}(y|x)\big) \\
&= \sum_{y\in\mathcal{A}} \pi^*(y|x)\big(\log\pi_{t+1}(y|x) - \log\pi_t(y|x)\big) - \pi_{t+1}(y|x)\big(\log\pi_{t+1}(y|x) - \log\pi_t(y|x)\big) \\
&= \sum_{y\in\mathcal{A}} \pi^*(y|x)\big(\log\pi^*(y|x) - \log\pi_t(y|x)\big) + \sum_{y\in\mathcal{A}} \pi^*(y|x)\big(\log\pi_{t+1}(y|x) - \log\pi^*(y|x)\big) \\
&\quad - \sum_{y\in\mathcal{A}} \pi_{t+1}(y|x)\big(\log\pi_{t+1}(y|x) - \log\pi_t(y|x)\big) \\
&= \mathrm{KL}\big(\pi^*(\cdot|x)\|\pi_t(\cdot|x)\big) - \mathrm{KL}\big(\pi^*(\cdot|x)\|\pi_{t+1}(\cdot|x)\big) - \mathrm{KL}\big(\pi_{t+1}(\cdot|x)\|\pi_t(\cdot|x)\big), \tag{F.7}
\end{aligned}$$

where the first equation holds due to (F.6) and the second equation holds due to $\sum_{y\in\mathcal{A}}\big(\pi^*(y|x) - \pi_{t+1}(y|x)\big) = 0$. Consequently, we have

$$\begin{aligned}
&\mathbb{E}_{y\sim\pi^*(\cdot|x)}\big[\widehat{D}_t(x,y)\big] - \mathbb{E}_{y\sim\pi_t(\cdot|x)}\big[\widehat{D}_t(x,y)\big] \\
&= \sum_{y\in\mathcal{A}} \widehat{D}_t(x,y)\big(\pi^*(y|x) - \pi_t(y|x)\big) \\
&= \sum_{y\in\mathcal{A}} \widehat{D}_t(x,y)\big(\pi^*(y|x) - \pi_{t+1}(y|x)\big) + \sum_{y\in\mathcal{A}} \widehat{D}_t(x,y)\big(\pi_{t+1}(y|x) - \pi_t(y|x)\big) \\
&\le \sum_{y\in\mathcal{A}} \widehat{D}_t(x,y)\big(\pi^*(y|x) - \pi_{t+1}(y|x)\big) + 2\big\|\pi_{t+1}(\cdot|x) - \pi_t(\cdot|x)\big\|_1 \\
&= \eta^{-1}\Big(\mathrm{KL}\big(\pi^*(\cdot|x)\|\pi_t(\cdot|x)\big) - \mathrm{KL}\big(\pi^*(\cdot|x)\|\pi_{t+1}(\cdot|x)\big) - \mathrm{KL}\big(\pi_{t+1}(\cdot|x)\|\pi_t(\cdot|x)\big)\Big) \\
&\quad + 2\big\|\pi_{t+1}(\cdot|x) - \pi_t(\cdot|x)\big\|_1 \\
&\le \eta^{-1}\Big(\mathrm{KL}\big(\pi^*(\cdot|x)\|\pi_t(\cdot|x)\big) - \mathrm{KL}\big(\pi^*(\cdot|x)\|\pi_{t+1}(\cdot|x)\big)\Big) \\
&\quad + 2\big\|\pi_{t+1}(\cdot|x) - \pi_t(\cdot|x)\big\|_1 - \frac{\big\|\pi_{t+1}(\cdot|x) - \pi_t(\cdot|x)\big\|_1^2}{2\eta} \\
&\le 2\eta + \eta^{-1}\Big(\mathrm{KL}\big(\pi^*(\cdot|x)\|\pi_t(\cdot|x)\big) - \mathrm{KL}\big(\pi^*(\cdot|x)\|\pi_{t+1}(\cdot|x)\big)\Big), \tag{F.8}
\end{aligned}$$

where the first inequality holds due to the fact that $0 \le \widehat{D}_t(x,y) \le 2$, the second inequality holds due to Pinsker's inequality and the last inequality holds due to the fact that $ax - bx^2 \le a^2/4b$. Finally, taking expectation over $x \sim \mathcal{D}$ finishes the proof. $\square$

## G Auxiliary Lemmas

**Lemma G.1** (Lemma A.2, Shalev-Shwartz & Ben-David, 2014). Let $a \ge 1$ and $b \ge 0$, then $x \ge 4a\log(2a) + 2b$ results in $x \ge a\log x + b$.

**Lemma G.2** (Theorem 1, Abbasi-Yadkori et al., 2011). *Let $\{\mathcal{F}_t\}_{t=0}^{\infty}$ be a filtration. Let $\{\epsilon_t\}_{t=1}^{\infty}$ be a real-valued stochastic process such that $\epsilon_t$ is $\mathcal{F}_t$-measurable and $\epsilon_t$ is conditionally $R$-sub-Gaussian for some $R \geq 0$. Let $\{\phi_t\}_{t=1}^{\infty}$ be an $\mathbb{R}^d$-valued stochastic process such that $\phi_t$ is $\mathcal{F}_{t-1}$ measurable and $\|\phi_t\|_2 \leq L$ for all $t$. For any $t \geq 0$, define $\mathbf{U}_t = \lambda \mathbf{I} + \sum_{i=1}^{t} \phi_i \phi_i^{\top}$. Then for any $\delta > 0$, with probability at least $1 - \delta$, for all $t \geq 0$, we have*

$$\left\| \sum_{i=1}^{t} \phi_i \epsilon_i \right\|_{\mathbf{U}_t^{-1}}^2 \leq 2R^2 \log \left( \frac{\sqrt{\det(\mathbf{U}_t)}}{\sqrt{\det(\mathbf{U}_0)}\delta} \right).$$

**Lemma G.3** (Lemma 11, Abbasi-Yadkori et al. 2011). *Let $\{\phi_i\}_{i=1}^{t}$ be a sequence in $\mathbb{R}^d$, define $\mathbf{U}_i = \lambda \mathbf{I} + \sum_{i=1}^{t} \phi_i \phi_i^{\top}$, then*

$$\sum_{i=1}^{t} \min \left\{ 1, \|\phi_i\|_{\mathbf{U}_{i-1}^{-1}}^2 \right\} \leq 2d \log \left( \frac{\lambda d + tL^2}{\lambda d} \right).$$

The following auxiliary lemma and its corollary are useful

**Lemma G.4** (Lemma A.2, Shalev-Shwartz & Ben-David 2014). *Let $a \geq 1$ and $b > 0$. Then $x \geq 4a \log(2a) + 2b$ yields $x \geq a \log(x) + b$.*

**Lemma G.5** (Lemma C.7, Zhang et al., 2023). *Suppose sequence $\{\mathbf{x}_i\}_{i=1}^{t} \subset \mathbb{R}^d$ and for any $i \leq t$, $\|\mathbf{x}_i\|_2 \leq L$. For any index subset $\mathcal{C} \subseteq [t]$, define $\mathbf{U} = \lambda \mathbf{I} + \sum_{i \in \mathcal{C}} \mathbf{x}_i \mathbf{x}_i^{\top}$ for some $\lambda > 0$, then $\det(\mathbf{U}) \leq (\lambda + |\mathcal{C}|L^2/d)^d$.*

**Lemma G.6** (Azuma–Hoeffding inequality, Cesa-Bianchi & Lugosi 2006). *Let $\{x_i\}_{i=1}^{n}$ be a martingale difference sequence with respect to a filtration $\{\mathcal{G}_i\}$ satisfying $|x_i| \leq M$ for some constant $M$, $x_i$ is $\mathcal{G}_{i+1}$-measurable, $\mathbb{E}[x_i|\mathcal{G}_i] = 0$. Then for any $0 < \delta < 1$, with probability at least $1 - \delta$, we have*

$$\sum_{i=1}^{n} x_i \leq M \sqrt{2n \log(1/\delta)}.$$

**Lemma G.7** (Lemma 11 in Abbasi-Yadkori et al. (2011)). *Let $\{\phi_t\}_{t=1}^{+\infty}$ be a sequence in $\mathbb{R}^d$, $\mathbf{U}$ a $d \times d$ positive definite matrix and define $\mathbf{U}_t = \mathbf{U} + \sum_{i=1}^{t} \phi_i^{\top} \phi_i$. If $\|\phi_i\|_2 \leq L$ and $\lambda_{\min}(\mathbf{U}) \geq \max(1, L^2)$, then we have*

$$\sum_{i=1}^{t} \phi_i^{\top} (\mathbf{U}_{i-1})^{-1} \phi_i \leq 2 \log \left( \frac{\det \mathbf{U}_t}{\det \mathbf{U}} \right).$$

