# OpenReview forum: "Reinforcement Learning from Human Feedback with Active Queries"
_TMLR — Accepted by TMLR_

### Review · Reviewer_grAN · 2025-06-20

**Summary Of Contributions:**

This paper proposes a novel active learning variant of PPO and DPO (APPO and ADPO, respectively) as a sample-efficient way to perform reinforcement learning from human feedback (RLHF). The proposed method formulates the RLHF problem as a contextual dueling bandit problem, where an estimator for the reward gap between the two arms is trained with maximum likelihood. Importantly, the active learning algorithm only queries human preferences for data where the model exhibits high uncertainty.

After presenting the query-efficient Active Proximal Policy Optimization (APPO) algorithm, the paper presents some theoretical analysis about the efficiency (level of regret and query complexity) of the algorithm. It then presents the more practical Active Direct Preference Optimization (ADPO) algorithm before conducting experiments with ADPO.

Two base models (Zephyr-Beta-SFT and Zephyr-Gemma-SFT) are trained with DPO and ADPO on the same human instruction datasets (Ultrachat-200k and deita-10k-v0-sft, respectively) and compared on multiple tasks (ARC, TruthfulQA, HellaSwag, MT-Bench, and Alpaca Eval). Results show that the ADPO-trained models perform similarly (if not better) than the DPO-trained variants at a fraction (¼ to ½) of the number of queries. Ablation studies also show the importance of the pseudo-labels and the effect of the confidence threshold for the learner to make a human label query.

**Audience:**

Yes

**Broader Impact Concerns:**

The current broader impact statement in the paper is good.

**Claims And Evidence:**

Yes

**Requested Changes:**

- Add some sort of compute or time requirement for each model presented in Table 1 & 2 to have a tradeoff discussion in the results.

- Explore lower threshold values in Table 4 (at least 1). Ideally, a figure with the X-axis being the threshold (with corresponding number of queries in parentheses) and the Y-axis being the average performance would be amazing, especially with very high and very low thresholds.

- The caption of Table 2 has a copy-pasted sentence from the caption of Table 1: “_Besides, ADPO only makes 16k queries … made by DPO_”. But Table 2 does not report the number of queries made. It would be nice to report the number queries made on MT-Bench and Alpaca Eval.

Typos:
- Under “Uncertainty-Aware Query Criterion” paragraph in Section 4: “_a pair of action requires y^1_t and y^2_t requires human-labelled preference._”.
- In Theorem 5.1: “_… in Algorithm 1, then with with probability at least…_”
- In Appendix B, Evaluation setup paragraph: “_For obejctive benchmarks_”

**Strengths And Weaknesses:**

## Strengths

- This paper provides both a theoretical analysis and a practical application of the proposed active learning algorithm.
- The paper is well structured and clearly motivated.
- Experiments are well presented with two models trained on instruction-tuned datasets and evaluated on well-recognized datasets.
- The work provides a data-efficient alternative to RLHF, making this process cheaper to run and more accessible.

## Weaknesses

**[W1]** The paper does not report the additional computational cost of using the proposed method in the experimental section. While active learning decreases the number of queries made (or samples used), the evaluation of the model, the potential query informativeness and other criteria probably adds computational costs. It would be nice to report such metrics to better visualize the tradeoff. Metrics like GPU hours or FLOPS could be used for that.

**[W2]** While the method relies on fewer samples, in practice, one does not know how many samples will be needed in advance. As such, active learning methods only make sense in a labeling process with human labelers in the loop, which is slower (though cheaper) than gathering a big dataset of labels all at once. For simplicity, the paper currently uses an already labeled dataset of a fixed size. It would be nice to discuss this limitation in the main text.

**[W3]** Finally, the ablation study of the confidence threshold currently covers values from 1.0 to 1.5, requiring from 13k to 21k samples, and all achieving better performance than the DPO baseline (Table 4). It would be very interesting to also report much slower thresholds and to identify the minimal threshold before the performance of the model collapses. What happens in practive if the thresholds is 0.0 and the learner never asks for a label? Recent work, such as “_Reinforcement Learning for Reasoning in Large Language Models with One Training Example_” suggest that only a very small amount of data is needed to achieve good reasoning performance. While these results are to be taken “with a grain of salt”, the work presented in this paper could potentially shed some light into these minimal training data paradigms.

---

> ### Author Response · Authors · 2025-08-06
>
> Thank you for your insightful and constructive feedback! We would like to answer your question as follows
>
> **W1 & RC1: Add some sort of compute or time requirement for each model presented in Table 1 & 2 to have a tradeoff discussion in the results**.
>
> **A1**: We thank the reviewer for the suggestion and we have added the time requirement in Appendix B. Specifically, under our experiment setup, the time required for DPO and ADPO is almost the same. This is primiary because that, in ADPO, the (almost) only computation extra to DPO is the computation of the uncertainty threshold and its comparison to the confidence threshold $\gamma$. Since uncertainty is computed via the difference of the predicted rewards of the two answers, and predicting the rewards themselves is a sub-routine of DPO, the data-filtering process in ADPO has almost no overhead at all. This lightweight nature constitutes one of the advantages of ADPO. We highlighted this at the end of Section 6 in our revision.
>
> **W2: Active learning methods only make sense with human labelers in the loop, which is slower (though cheaper) than gathering a big dataset of labels all at once.**
>
> **A2**: We thank the reviewer for the insightful comments and suggestion. We agree that when the labeler is human, human labeler in the loop might not be efficient as gathering a large amount of labels at once. Nevertheless, in some scenarios the preference labels are given by language models or verifiers. In this scenarios, calling these "labelers" in the loop does not imposes additional costs, and the cost accumulates by the number of calling. Under these scenarios, fewer sample is indeed more favorable. We thank the reviewer for understanding that this paper uses a labeled dataset primarily for simplicity. We have add discussion (footnote 8) to this limitation in our revision.
>
> **W3 & RC2**: Explore lower threshold values in Table 4 (at least 1). Ideally, a figure with the X-axis being the threshold (with corresponding number of queries in parentheses) and the Y-axis being the average performance would be amazing
>
> **A2**: We thank the reviewer for the suggestion. We selected several other values of $\gamma$ ranging from $0.2$ to $4.0$ and trained from Zephyr-Beta-SFT. We test the corresponding resulted checkpoints on objective benchmarks and plot the performances in Figure 3. The results show that the performance of ADPO matches DPO for a wide range of $\gamma$, indicating the robustness of ADPO against parameter selection
>
> **RC3: It would be nice to report the number queries made on MT-Bench and Alpaca Eval.**
>
> **A3**: Actually, the checkpoints for subjective benchmark evaluation is the exactly the same checkpoints for objective benchmark evaluation, which are obtained by training from Zephyr-Beta-SFT or Zephyr-Gemma-SFT on Ultrafeedback-binarized dataset. Therefore, they share the same numbers of queries. We have enphasized this in our revision.
>
> **Q4: Typos**
>
> **A4**: Thank you so much for pointing out these typos! We have fixed them in our revision.

---

> > ### Comment · Reviewer_grAN · 2025-08-15
> > **thanks for the clarifications**
> >
> > Dear authors,
> >
> > Thank you for taking the time to address all my suggestions and questions. It is especially interesting to see the results in Figure 3. In particular, we see that as you increase the threshold, the performance gets closer and closer to the original DPO performance. Also it shows that using fewer queries is beneficial (up to a certain point of course). I think this highlights an interesting learning dynamic regarding the quality and quantity of data needed for training.

---

> > > ### Author Response · Authors · 2025-08-18
> > >
> > > We sincerely thank the reviewer for acknowledging our rebuttal, the new results and revision. We are grateful for your thoughtful feedback and are pleased that you found our updated findings interesting.

---

### Review · Reviewer_n1tV · 2025-07-23

**Summary Of Contributions:**

This paper tackles the high cost of human (or proxy) preference queries in Direct Preference Optimization (DPO) by introducing Active DPO, which adds a lightweight “gate‑keeper” model to decide when to ask for a preference label. In standard DPO, every pair of model outputs
is sent to an expensive reward model—a human or large LLM—to obtain a preference label. Active DPO interposes a simple logistic regression classifier trained online: for each input, the base model proposes two outputs, the gate‑keeper predicts which is better and with what confidence, and only if that confidence falls below a threshold γ does the system defer to the oracle and add the true label to the classifier’s training data. By trusting the gate‑keeper whenever, the method auto‑labels “easy” cases and spares the oracle, while still querying on hard examples where its cost is justified. Experiments using a Zephyr‑based architecture show that Active DPO can cut total queries by roughly half with negligible loss in task accuracy. An ablation over different γ values (1.0, 1.3, 1.5) further illustrates the trade‑off between query volume and performance: higher thresholds yield fewer oracle calls but incur a small accuracy drop. In essence, Active DPO retains nearly all the benefits of full DPO at a fraction of the labeling expense by adaptively routing only uncertain cases to the costly reward model.

**Audience:**

Yes

**Claims And Evidence:**

Yes

**Requested Changes:**

No, I think paper has covered everything in detail. If you could explain few things in detail?
1. One thing thats bothering me, are we really using simple logistic regression or is the paper using NN based logistic regression?
2. Is reward model another LLM pre-trained on preference dataset or its human labeller who gets the prediction queries to label as per preference?

**Strengths And Weaknesses:**

Strengths :
The paper is exceptionally clear and well‑structured: concepts are introduced logically, and the training procedure for Active DPO is laid out step by step. Diagrams and pseudocode are easy to follow, and the appendix provides all necessary implementation details.

The ablation studies are thorough, quantifying how the confidence threshold γ affects both query volume and end‑task performance. This not only validates the core idea but also offers practical guidance for tuning.

Adding a simple, online‑trained logistic regression “gate‑keeper” represents a small yet highly effective extension to vanilla DPO—cutting human/LLM queries by roughly half with minimal loss in accuracy. The elegance of this minimal intervention makes it broadly applicable.

Experimental evaluation on the Zephyr backbone demonstrates real gains, and comparisons against both always‑query DPO and a passive baseline firmly establish the benefits of the active approach.

Weakness :
All experiments use the Zephyr architecture. It remains unclear how well Active DPO transfers to other larger LLMs (LLM with more params) or multi-model.

The approach relies on a fixed confidence cutoff. In reality, the optimal γ may drift over time or vary by data domain; a dynamic or adaptive threshold could offer further gains.

Logistic regression may struggle on more complex tasks where the decision boundary between outputs is highly nonlinear. Testing richer classifiers  could reveal whether the choice of gate‑keeper matters.

---

> ### Author Response · Authors · 2025-08-06
>
> Thank you for your insightful feedback! We answer your questions as follows
>
> **W1: All experiments use Zephyr architecture.**
>
> **A1**: Actually, Zephyr-Beta-SFT is trained from Mistral-7B and, while Zephyr-Gemma-SFT is trained from gemma-7b, which means that these two models have different architecture (Zephyr is the name given by the repo alignment handbook). Therefore, our experiments show that ADPO generalizes between different architectures. We acknowledge that due to computation resources, currently our experiments are restricted to 7B-LLMs and we leave exploring query-efficient methods for multi-modal LLM and larger LLM as a future direction.
>
> **W2: The approach relies on a fixed confidence cutoff.**
>
> **A2**: We thanks for the reviewer's insightful comments! While our current approach, which fixes the confidence threshold, already warrants satisfactory performance, we agree with the reviewer that a dynamic or adaptive threshold might results to a further performance gain and leave this as our future direction.
>
> **W3: Logistic regression may struggle with more complex tasks**
>
> **A3**: We thank the reviewer for the insightful comments! Since this work primarily focuses on improving the data-efficiency of DPO, which uses logistic regression to fit the dataset, we also consider logistic regression. We agree that logistic regression might struggle with more complex tasks and might require more complicated classifiers when potentially applying our methods to other scenarios.
>
> **RC1: Is this paper really using simple logistic regression or is the paper using NN based logistic regression?**
>
> **A1**: In this paper, the reward computation and label prediction follows the setup in standard direct preference optimization (DPO), which is NN based logistic regression. Specifically, let $\theta$ be the parameter of the "being-optimized" large language model and $\pi_{\theta}$ denotes the policy induced by such language model (i.e., the probability of outputing $y$ given prompt $x$ is $\pi_{\theta}(y|x)$. In DPO, given a prompt $x$ and a response $y$, the reward function is computed as $r_{\theta}(x,y) = \beta (\log \pi_{\theta}(y|x) - \log \pi_{\text{ref}}(y|x))$. Then, given two answers $y^1$ and $y^2$, the predicted label is given by $\sigma(r_{\theta}(x,y^1) - r_{\theta}(x,y^2))$, where $\sigma$ is the logistic function. Since $\theta$ is the parameter of a neural network, we are using an NN based logistic regression.
>
>
> **RC2: Is reward model another LLM pre-trained on preference dataset or its human labeller who gets the prediction queries to label as per preference?**
>
> **A2**: In this paper, we use a labeled dataset, namely Ultrafeedback-binarized for ease of conducting the experiment. Ultrafeedback-binarized contains about 60k prompts, where each prompt corresponds to two answers and the preference labels. In the experiment of this paper, the being-training LLM (which is also a reward model as formulated by DPO) predict the reward difference of these two answers. If the model is not confident with its prediction, then we use the label in the dataset for training. In real world scenarios, usually there are no labels and we need to call a true reward model (which is not the being-trained LLM) to provide the labels. This reward model might be any reward model depending on specific scenarios. This procedure might be expensive, which motivates the query-efficient approach proposed by this paper.

---

### Review · Reviewer_Ga6d · 2025-07-28

**Summary Of Contributions:**

This paper introduces a query-efficient framework for reinforcement learning from human feedback (RLHF) by incorporating active learning strategies. The authors formulate RLHF as a contextual dueling bandit problem and propose a theoretically grounded algorithm called Active Proximal Policy Optimization (APPO), which achieves instance-dependent regret bounds independent of the action space size. Furthermore, the authors develop a practical instantiation of APPO named Active Direct Preference Optimization (ADPO), which integrates pseudo-labeling into a DPO-style training pipeline. Experimental results on multiple large language models (LLMs), including Zephyr-7b variants, across both objective (e.g., ARC, TruthfulQA, HellaSwag) and subjective benchmarks (e.g., MT-Bench, AlpacaEval 2.0), demonstrate that ADPO achieves performance comparable to or better than standard DPO while requiring significantly fewer preference queries.

**Audience:**

Yes

**Broader Impact Concerns:**

This work addresses an important bottleneck in RLHF by reducing reliance on human-labeled preferences, which has direct implications for scalability and accessibility of LLM alignment.

**Claims And Evidence:**

Yes

**Requested Changes:**

Same as the weakness above.

**Strengths And Weaknesses:**

## Strengths
1. This paper tackles a highly relevant and practical problem in LLM alignment, i.e., the high cost of collecting human preference data.

2. APPO is rigorously analyzed with solid theoretical guarantees, featuring regret and query complexity bounds that improve over prior work by removing dependency on the action space.

3. ADPO introduces a compelling pseudo-labeling mechanism for training under uncertainty, which is empirically shown to be critical for data efficiency.

4. Experimental results are comprehensive, covering both objective and subjective evaluation settings, and including ablation studies on pseudo-labeling and uncertainty thresholds.
5. The work contributes to bridging theoretical insights with practical LLM training pipelines.

## Weakness
1. The pseudo-labeling mechanism, while empirically justified, lacks a quantitative analysis of its error rates or possible accumulation of bias during training.
2. Comparisons are limited to DPO; other related baselines such as PPO with active querying, AURORA, or recent active RLHF strategies are not included
3. The choice of uncertainty threshold γ is empirically tuned but lacks adaptive mechanisms or principled justification

---

> ### Author Response · Authors · 2025-08-06
>
> Thank you for your insightful and constructive feedback! We answer your questions as follows
>
> **W1: The pseudo-labeling mechanism lacks a quantitative analysis of its error rates or possible accumulation of bias during training.**
>
> **A1**: We thank the reviewer for the suggestion and added a paragraph in Section 8.1 to discuss the quality of the pseudo-labels. Specifically, we count the total number of predicted labels and the number of correctly predicted labels and compute the correct rate. The results are compiled in Table 4 and we also present a copy here. The results show that the correct rate remains above 75% quickly after the start of the training, which means that pseudo-labels can serve as meaningful guidance.
>
>
> | Steps            | 50    | 100   | 200   | 300   | 450   | 600   | 750   | 950   |
> | ---------------- | ----- | ----- | ----- | ----- | ----- | ----- | ----- | ----- |
> | Predicted labels | 24    | 952   | 4904  | 9840  | 17448 | 25432 | 33232 | 43904 |
> | Correct Labels          | 16    | 808   | 3936  | 7792  | 13432 | 19392 | 25384 | 33488 |
> | Error Rate  (%)      | 66.67 | 84.87 | 80.26 | 79.19 | 76.98 | 76.25 | 76.38 | 76.28 |
>
>
> **W2: Comparisons are limited to DPO, other related baselines are not included**
>
> **A2**: We thank the reviewer for this suggestion! We consider AE-DPO introduced in [1] and the Active Preference Learning (denoted by APL for simplicity) introduced in [2] as our baseline. These two methods also consider Direct Preference Optimization as the backbone but uses different filtering criteria (and also no pseudo-labeling) compared to ADPO. We report their performances on both models and all benchmarks and refer the reviewer to Table 1 and Table 2 in our revised paper. We see that ADPO notably performs better than these two baselines. Furthermore, we would like to highlight that AE-DPO requires multiple forward computation to estimate uncertainty, which is much slower than our approach. These results indicate the superiority of ADPO over the baseline methods.
>
> **W3: The choice of uncertainty threshold $\gamma$ is empirically tuned but lacks adaptive mechanisms or principled justification**
>
> **A3**: This is a good question. Actually, $\gamma$ can be determined by a grid search when training on a small portion of the dataset. Then, the value of $\gamma$ yielding the best best performance could be selected as the final choice of $\gamma$. Furthermore, as we presented in our ablation study, or more specifically, Figure 5, the performance of ADPO matches the performance of DPO when $\gamma$ ranges from 0.5 to 4.0, which means $\gamma$ does not need to be carefully tuned to ensure a good performance. We have added the above discussion to our revised paper (the paragraph between equation (6.4) and (6.5)).
>
> ---
>
> References
>
> [1] Mehta, Viraj, et al. "Sample efficient reinforcement learning from human feedback via active exploration." arXiv preprint arXiv:2312.00267 (2023).
>
> [2] Muldrew, William, et al. "Active Preference Learning for Large Language Models." arXiv preprint arXiv:2402.08114 (2024).

---

### Author Response · Authors · 2025-08-06

Dear Editor and Reviewers,

Thank you for your time in providing valuable feedback on our manuscript. We have incorporated all your suggestions in our revision, with the revised texts highlighted in blue. To summarize, besides typos and clarifications, we made the following revisions

1. We added a quantitative analysis of the error rate of pseudo-labels in Section 8.1 and the results are compiled in Table 4, which empirically verified the quality of pseudo-labels
2. We added the comparison to two concurrent DPO-based active-learning methods, whose results are also summarized in Table 1 and Table 2. These results show the superiority of ADPO over these baselines
3. We explored a wider range of $\gamma$ and compiled the results in Figure 3. These results provide a more comprehensive understanding of how ADPO performs with different $\gamma$.

We are more than happy to provide more clarifications and revisions if needed.


Best,

Authors

---

### Decision · Action_Editor_WDpM · 2025-08-20

**Recommendation:** Accept as is

**Additional Comments:**

The reviewers agree this submission makes a solid and interesting contribution to RLHF by studying how to actively seek for preference feedback where it is needed the most (the generative model has high uncertainty). Due to reviewers consensus and my own good impression about the paper and the potential interest it may draw from RL and LLM communities, I am recommending to accept the paper as is together with a feature certification.

**Audience:**

Yes

**Audience Explanation:**

Yes, I believe this paper can interest both the RL/bandits community and the LLM community,

**Claims And Evidence:**

Yes

**Claims Explanation:**

The paper delivers on the claim made in the abstract and introduction by providing a methodology for active queries in a dueling contextual bandit formulation of RLHF. The paper contributes algorithmic extensions to PPO and DPO for preference feedback, for which the corresponding regret is analyzed, as well as practical implementations corroborated in a set of experiments about reasoning and text generation.